# A mussel-inspired film for adhesion to wet buccal tissue and efficient buccal drug delivery

Shanshan Hu [1,2,4], Xibo Pei[1,4], Lunliang Duan [3], Zhou Zhu [1], Yanhua Liu [1], Junyu Chen [1], Tao Chen [2], Ping Ji [2], Qianbing Wan [1✉] & Jian Wang [1✉]

Administration of drugs via the buccal route has attracted much attention in recent years. However, developing systems with satisfactory adhesion under wet conditions and adequate drug bioavailability still remains a challenge. Here, we propose a mussel-inspired mucoadhesive film. Ex vivo models show that this film can achieve strong adhesion to wet buccal tissues (up to 38.72 ± 10.94 kPa). We also demonstrate that the adhesion mechanism of this film relies on both physical association and covalent bonding between the film and mucus. Additionally, the film with incorporated polydopamine nanoparticles shows superior advantages for transport across the mucosal barrier, with improved drug bioavailability (~3.5-fold greater than observed with oral delivery) and therapeutic efficacy in oral mucositis models (~6.0-fold improvement in wound closure at day 5 compared with that observed with no treatment). We anticipate that this platform might aid the development of tissue adhesives and inspire the design of nanoparticle-based buccal delivery systems.

[1] State Key Laboratory of Oral Diseases, National Clinical Research Center for Oral Diseases, Department of Prosthodontics, West China Hospital of Stomatology, Sichuan University, Chengdu, China. [2] Chongqing Key Laboratory of Oral Diseases and Biomedical Sciences, Chongqing Municipal Key Laboratory of Oral Biomedical Engineering of Higher Education, Stomatological Hospital of Chongqing Medical University, Chongqing, China. [3] National Engineering Research Center for Inland Waterway Regulation, Chongqing Jiaotong University, Chongqing, China. [4] These authors contributed equally: Shanshan Hu, Xibo Pei. ✉email: champion@scu.edu.cn; ferowang@hotmail.com

Every day, millions of patients worldwide are subjected to undesirable side effects caused by high dosages of drugs administered via the oral route[1] or a fear of injections[2]. An estimated 80% of patients suffer from severe needle phobia[2]. Buccal drug delivery, which is non-invasive, painless and convenient, offers superb advantages over the oral and parenteral formulations[1]. By avoiding enzymatic degradation in the gastro-intestinal tract as well as first-pass hepatic metabolism[3,4], delivery of drugs via the buccal mucosa would markedly improve patient experiences and disease outcomes. However, there are also many challenges associated with this route that have limited its further implementation. The main challenge related to the buccal route is residence time[5]. To ensure optimal drug efficiency, buccal drug delivery formulations need to maintain intimate contact with the oral mucosa for a long period of time. However, flushes of saliva, swallowing, and mouth movement can all influence the residence time of formulations in the buccal cavity[3,4]. Therefore, achieving strong adhesion under the wet conditions of the oral cavity is challenging. In addition, the buccal administration route has substantial transport barriers. Drugs must diffuse the mucus layer that covers the surface of the oral mucosa and be transported across the epithelial layer to be absorbed[6,7]. Unfortunately, despite the overwhelming clinical need for optimal buccal drug delivery, few formulations that exhibit excellent properties to overcome these limitations have been reported.

Marine mussels, which are well known for their remarkable underwater adhesion ability, have attracted widespread attention and are a potential source of an ideal tissue adhesive in the biological field[8,9]. The rapid and robust adhesion of mussels could be attributed to the presence of the mussel adhesive proteins, which are abundant in the catecholic amino acid 3,4-dihydroxyphenylalanine (DOPA)[8,9]. The catechol group of DOPA has an excellent maneuverability during crosslinking because it forms either covalent or noncovalent bonds. First, it can form non-covalent complexes, as in metal bidentate coordination and hydrogen bonding. In addition, the catechol groups oxidize easily to form o-quinone in oxidative or alkaline environments. The oxidized o-quinone is highly susceptible to forming covalent bonds with nucleophiles such as thiols and amines of proteins on the tissue surface via Michael addition or Schiff base reactions. In addition, o-quinone can also form di-dopa crosslinks via phenol radical coupling[8–10]. Inspired by this functional group from mussels, enormous efforts has been devoted to the development of catechol-functionalized adhesives via modifications of a variety of mucoadhesive polymers, including catechol-modified poly(ethylene glycol) (PEG)[11,12], chitosan[5,13], hyaluronic acid (HA)[14], alginate[15], etc. However, most of those studies focused on tissue adhesives for skin, and reports on formulations that exhibit excellent mucoadhesive properties in the wet oral environment and how they interact with the oral mucosa are still very limited. Although Xu et al.[5] reported a catechol-chitosan mucoadhesive hydrogel for buccal drug delivery, they did not investigate the ability of the drug to be transported across the epithelial barrier, and their hydrogel provided sustained drug release for only 3 h. Therefore, attempts should also be made to develop drug carriers that could be transported across the epithelial barriers with a controlled and prolonged drug release profile.

Recently, nanoparticles (NPs) have shown great promise for improved transport through the mucus barrier and can be tuned to support controlled or sustained release behavior[16–18]. Among various strategies, surface modification of NPs with PEG has emerged as a popular strategy to enhance the mucus-penetrating ability of NPs[19–21]. Nevertheless, the hydrophilic and neutral surface properties of PEG may serve as a barrier for further cellular uptake of NPs[22]. Therefore, strategies that could achieve both excellent mucus-penetrating ability and cellular uptake

across the epithelial barriers are also in high demanded. To meet these challenges, the use of a mussel-inspired tissue adhesive combined with NPs as drug carriers might be a good strategy to overcome the limitations associated with the buccal administration route.

Herein, we propose a buccal tissue adhesive in the form of a tunable thin film made from a combination of the mucoadhesive polymer poly(vinyl alcohol) (PVA) and the mussel adhesive protein DOPA (PVA-DOPA film). Ex vivo porcine and in vivo rat models show that the film can achieve strong adhesion and good mechanical matching with wet buccal tissues. We also demonstrate that the films exhibit tunable mucoadhesion strength and erosion rates in proportion to the amount of DOPA. The adhesion mechanism of this film relies on both physical association and covalent bonding between the film and mucus. Then, we adopt three kinds of polymers, PEG, PVA, polydopamine (PDA), to assemble core-shell poly(lactic-co-glycolic acid) (PLGA) NPs with different surface coatings (Fig. 1a) and incorporated them into the PVA-DOPA film to form a combined buccal drug delivery system (PVA-DOPA@NPs film) (NPs refer to PLGA, PLGA-PEG, PLGA-PVA or PLGA-PDA NPs), as shown schematically in Fig. 1b. Through systematic evaluation, we observe that the PDA modified NPs exhibit improved mucus-penetrating ability and cellular uptake (Fig. 1c). In addition, the buccally administered PVA-DOPA film incorporated with drug-loaded PLGA-PDA NPs show the best bioavailability and superior therapeutic efficacy for oral mucositis. Therefore, we present an integrated buccal drug delivery system in which NPs show a controlled drug release profile and improved mucus-penetrating ability and cellular uptake, and the shielding film shows a prolonged residence time and improved mucoadhesion.

## Results

**Preparation and characterization of the PVA-DOPA mucoadhesive film.** First, PVA-DOPA polymers were synthesized by modifying PVA with the mussel adhesive protein DOPA. The formation of PVA-DOPA1-6 (according to the content of DOPA added, refer to the Materials section) was confirmed by Fourier transform infrared (FTIR), ultraviolet-visible (UV-vis), and $^1$H-nuclear magnetic resonance ($^1$H-NMR) spectra. The existence of a vibration absorption peak (1734 cm$^{-1}$) for the C=O bond proved the successful synthesis of PVA-DOPA polymers (Supplementary Fig. 1a). Furthermore, a shift of hydroxy groups (-OH) from 3254 cm$^{-1}$ for pure PVA to 3277 cm$^{-1}$ was observed with the addition of DOPA, suggesting the formation of hydrogen-bonding interactions between the PVA matrix and catechol[23]. The absorption peaks of catechol at 280 nm in the UV-vis spectra (Fig. 2a) and the peaks in the aromatic regions ($\delta = 6.62, 6.71, 6.78$) of the $^1$H-NMR spectra (Fig. 2b) also verified the conjugation of DOPA to PVA chains[24,25], and the height of the absorption peaks was proportional to the amount of DOPA added. Therefore, the degree of substitution of catechol in PVA-DOPA conjugates could be calculated by comparing the peak area of the phenyl group in catechol ($\delta = 6.62, 6.71, 6.78$) relative to that of the methylene group in the PVA chains ($\delta = 1.5$)[24,25]. As shown in Supplementary Table 1, the ratios of catechol conjugated to the PVA backbone were ~4.7–64.6%. Besides, the catechol content was confirmed by the UV–vis spectroscopy, measuring absorbance at 280 nm, and quantitative measurement was performed with a DOPA standard curve[26,27]. As shown in Supplementary Table 1, the mass fraction of catechol/PVA-DOPA ranged from 16.0 wt% to 72.0 wt%. Therefore, the catechol groups could be characterized and quantified by the UV-vis and $^1$H-NMR spectroscopy. Photographs of the ethyl cellulose protective cap, the lyophilized PVA-DOPA film and the film after

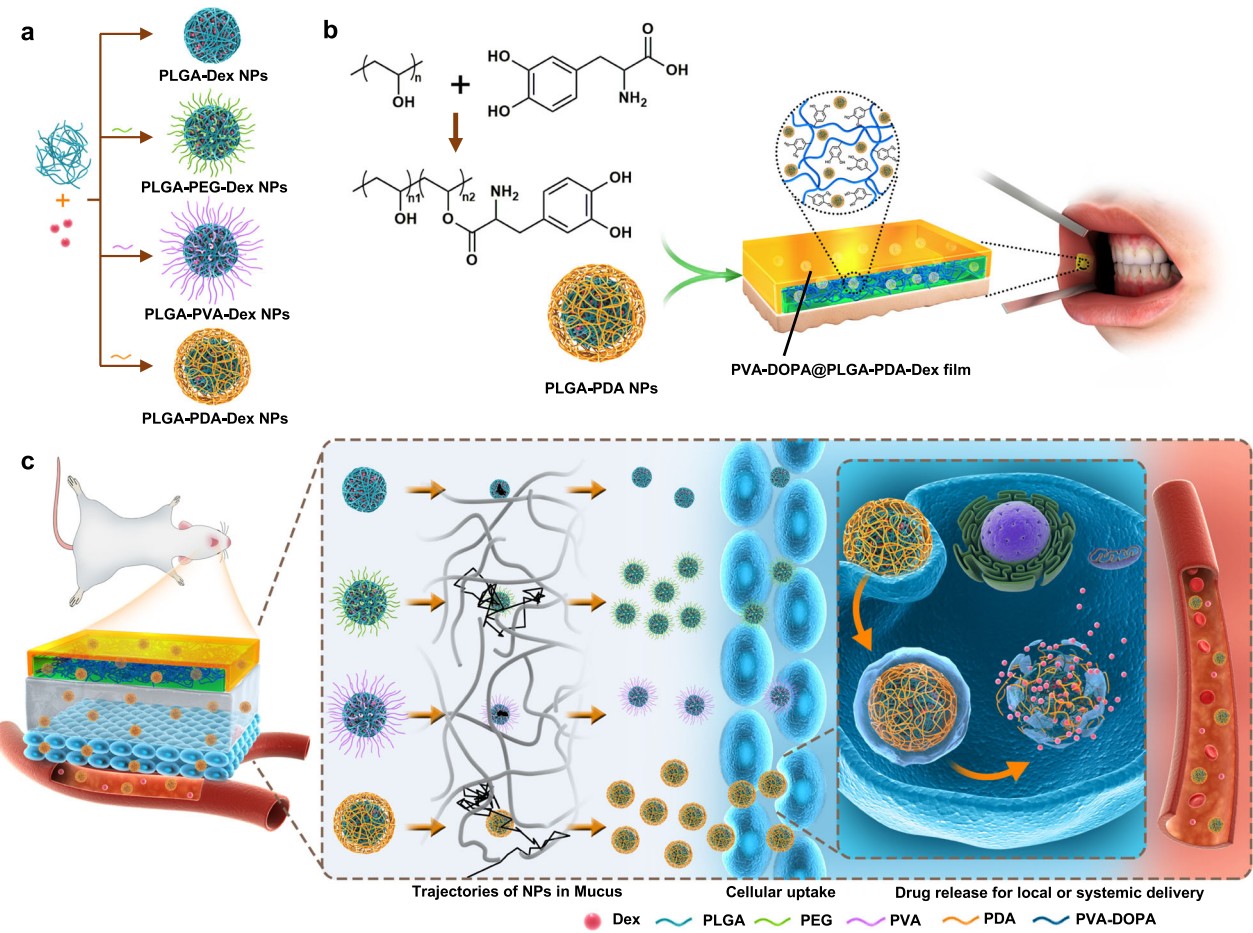

**Fig. 1 Synthesis and biomedical application of PVA-DOPA@NPs-Dex mucoadhesive film. a** Illustrations displaying the method used to assemble core-shell PLGA NPs with different surface modifications. **b** Schematic presentation of the fabrication of the PVA-DOPA@NPs-Dex film with enhanced mucoadhesion for buccal drug delivery. **c** Schematic diagram of the application of PVA-DOPA@NPs-Dex film to the rat buccal mucosa and the process by which the NPs to sequentially permeate the mucus layer and epithelial cells. PDA-coated PLGA NPs could overcome both barriers rapidly and subsequently release drugs for local or systemic delivery. NPs: nanoparticles, Dex: dexamethasone, PLGA: poly(lactic-co-glycolic acid), PEG: poly(ethylene glycol), PVA: poly(vinyl alcohol), PDA: polydopamine, DOPA: 3,4-dihydroxy-D-phenylalanine.

hydration are presented in Supplementary Fig. 1b-e. In addition, the thickness and surface pH of different PVA-DOPA samples are presented in Supplementary Table 2. As shown, the uniform thickness ensured accurate dosing of the prepared film formulations, and the near neutral surface pH indicated that there would be no potential irritation to the mucosa.

As it has been reported that the storage modulus ($G'$) is directly related to the crosslinking density and stiffness of the network[28,29], the results of rheological studies (Fig. 2c) suggested decreased crosslinking density of the networks with increasing DOPA content. In addition, since the increased loss modulus ($G''$) (Fig. 2d) indicated elevated viscous dissipation and the presence of reversible bonds in the hydrogel network[29–32], it could also be speculated that noncovalent crosslinking of the PVA-DOPA film occurred in the present study. Furthermore, the tensile strength and strain testing demonstrated good breakability and flexibility behavior of the films (Supplementary Fig. 1f-h) and indicated their ability to match the outline of the buccal cavity well after application, as they were capable of mechanically matching soft tissues[33]. Swelling assessment is essential to understand the mucoadhesive properties of the film and the release rate of drugs incorporated in the films[34,35]. All the prepared PVA-DOPA films showed sufficient hydration to form adhesions with the buccal mucosa, and the rate of hydration was rapid during the initial phase of ~2 h (Supplementary Fig. 1i).

However, it is not wise to compare the hydration rate of each PVA-DOPA film since the extent of erosion clearly increased progressively with the DOPA content (Fig. 2e). Therefore, the hydration rates of PVA-DOPA5 and PVA-DOPA6 decreased rapidly after 4 h due to their higher erosion rates (Fig. 2e). The reduction in the crosslinking density of the film likely increased the rate of degradation of the PVA-DOPA film with a high content of DOPA. Hence, the erosion rate of the film can be controlled over time periods from several hours to several days by tuning its composition, and this DOPA-related tunable erosion rate of the PVA-DOPA films also makes them applicable to various kinds of diseases that demand different dosage intervals.

**Adhesion performances of the PVA-DOPA mucoadhesive film.** The mucoadhesive properties of the PVA-DOPA films were determined in terms of residence time and mucoadhesion strength on freshly excised porcine buccal mucosa[5,36]. Two methods, the flow-through method and rotating disc method (Supplementary Fig. 2a, b)[5,37], were adopted to investigate the in vitro residence time, which is useful to evaluate whether the film can maintain its adhesion to the buccal mucosa surface for a sufficient time to ensure drug permeation. We evaluated the residence time of PVA-DOPA films by recording the number of

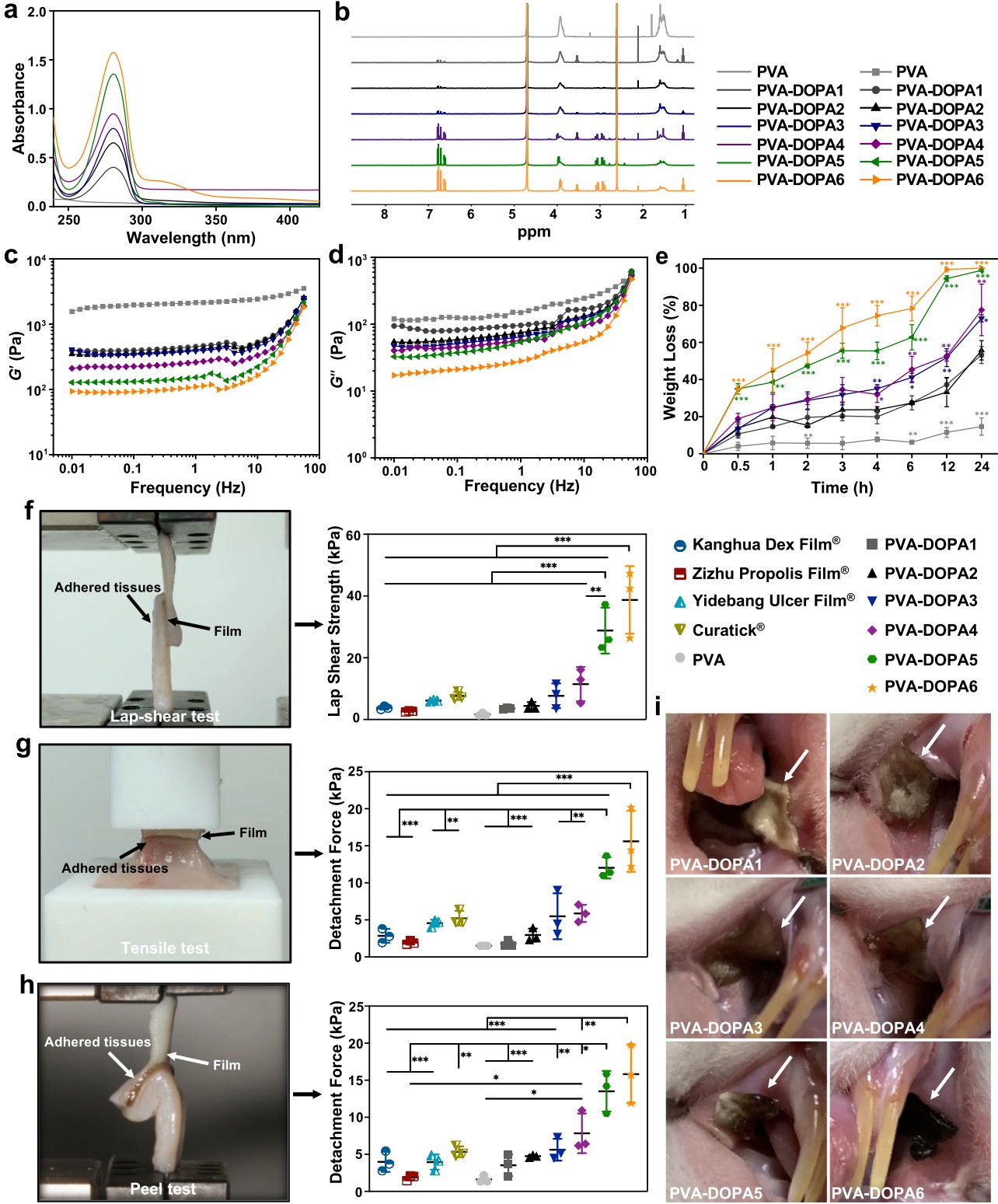

remaining films adhered to the porcine buccal mucosa. All unmodified PVA films detached from the mucosal surface within 1 h, showing that PVA alone provides only weak adhesion to the buccal mucosa (Supplementary Fig. 2c, d). In contrast, PVA-DOPA films adhered to the buccal tissue for a much longer time, especially PVA-DOPA5 and PVA-DOPA6, and nearly all films still adhered to the buccal tissue by the end of 8 h (data were only recorded up to 8 h due to the rapid erosion rate of the films).

Then, we conducted three types of mechanical tests to evaluate adhesion strength: shear strength was evaluated with lap-shear tests (Fig. 2f, Supplementary Movies 1 and 2), tensile strength was evaluated with tensile tests (Fig. 2g and Supplementary Movie 3), and interfacial toughness was evaluated with peel tests (Fig. 2h and Supplementary Movie 4). The PVA-DOPA film can establish tough and strong mucoadhesion with the wet porcine buccal mucosa upon contact for 10 s (Fig. 2f–h). Moreover, the mucoadhesion strength increased with the amount of DOPA,

**Fig. 2 Characterization and adhesion strength of PVA-DOPA films. a** UV-vis absorbance spectra of PVA-DOPA polymers with different amounts of DOPA. PVA: poly(vinyl alcohol), DOPA: 3,4-dihydroxy-D-phenylalanine. **b** $^1$H-NMR spectra of PVA-DOPA polymers with different DOPA contents. **c** Storage modulus (G′) of films with different DOPA contents. **d** Loss modulus (G′′) of films with different DOPA contents. **e** Erosion rates of films with different DOPA contents as a function of time. $n = 3$ independent samples per group; *$P < 0.05$; **$P < 0.01$; ***$P < 0.001$ vs PVA-DOPA1 group. **f** Comparison of the shear strength of PVA-DOPA films and various commercially available ulcer films by lap-shear tests. $n = 3$ independent samples per group; **$P = 0.004$; ***$P < 0.001$. **g** Comparison of the tensile strength of PVA-DOPA films and various commercially available ulcer films by tensile tests. $n = 3$ independent samples per group; **$P < 0.01$; ***$P < 0.001$. **h** Comparison of the interfacial toughness of PVA-DOPA films and various commercially available ulcer films by peel tests. $n = 3$ independent samples per group; *$P < 0.05$; **$P < 0.01$; ***$P < 0.001$. **i** In vivo mucoadhesion of PVA-DOPA films with different DOPA contents after 4 h. White arrow: PVA-DOPA films. $n = 3$ animals per group. All data are Mean ± S.D. Statistics was calculated by one-way ANOVA followed by Tukey's post-test. Exact $P$ values are given in the Source Data file. Source data are provided as a Source data file.

which confirmed that catechol groups can effectively increase the mucoadhesion strength. When compared with existing commercially available ulcer films, including Kanghua Dex Film® (containing vitamin B, dexamethasone, PVA, etc.) (China), Zizhu Propolis Film® (containing propolis cream, glycerin, PVA, povidone, etc.) (China), Yidebang Ulcer Film® (containing chitosan, PVA, etc.) (China), and Curatick® (containing glycerin, PEG, etc.) (Korea), we found that PVA-DOPA1-4 films exhibited adhesion performance on wet tissues that was comparable to or better than that of the four existing commercial films, while PVA-DOPA5 and PVA-DOPA6 films showed interfacial toughness, shear and tensile strength significantly better than those of existing commercially available films (Fig. 2f–h) ($P < 0.01$). Further in vivo tests also demonstrated that different PVA-DOPA films could maintain good contact with the mucosal tissue of Sprague-Dawley rats (Fig. 2i) after 4 h. In addition, as depicted in Supplementary Fig. 2e, the PVA-DOPA films also showed self-healing abilities after hydration, which indicated its potential to autonomously and rapidly heal after breakage in the oral environment.

**Exploration of the interaction mechanisms between the PVA-DOPA film and mucus.** When a suspension of mucin particles was mixed with polymers, the mucin particles aggregated to form larger sizes if the polymer had a strong affinity for them[13,38]. Therefore, the size and turbidity of the PVA-DOPA polymer after mixing with mucin were evaluated. As shown in Fig. 3a and Supplementary Fig. 3a, all films had a high affinity for mucin particles, and the extent of the change in particle size and turbidity was proportional to the ratio of DOPA. Moreover, the resultant zeta potential of mucin shifted to a higher negative value as a function of time, also confirming the adsorption of PVA or PVA-DOPA molecules to the mucin particles and indicating that the aggregation tendency depended on the content of DOPA (Fig. 3b).

To further elucidate the adhesion mechanism of the film, several spectral analyses were performed in an attempt to understand the molecular interactions that occur between the mucosal surfaces and the film. First, the catechol-mediated covalent reaction between different PVA-DOPA and mucin was monitored by means of UV-vis spectroscopy[13,39,40]. As shown in Fig. 3c and Supplementary Fig. 3b, there was no obvious difference between mucin and PVA-mucin at different concentrations. However, a slight shift could be observed in the three kinds of PVA-DOPA-Mucin complexes. In addition, the absorption spectra of PVA-DOPA-Mucin complexes were also different from those of PVA-DOPA or mucin alone, and the difference was lager with an increased concentration of PVA-DOPA (Supplementary Fig. 3b), indicating the covalent reaction of catechol with mucin[13,39,40]. We also investigated the interactions between mucin and PVA or PVA-DOPA using FTIR. The pure mucin exhibited amide I and amide II peak positions at 1645 and 1552 cm$^{-1}$, respectively (Fig. 3d), and there was no significant

difference between PVA and the PVA-Mucin complex at the amide I and II bands. However, for the three kinds of PVA-DOPA, amide II shifted to 1528 cm$^{-1}$ after reacting with mucin suspension, suggesting the covalent conjugation of catechol with mucin[41]. The $^1$H-NMR spectra of different kinds of PVA-DOPA before and after mixing with mucin also verified the covalent crosslinking between the oxidized catechol in DOPA and cysteine-rich glycoprotein domains in mucin (Supplementary Fig. 3c)[24]. In addition, thermodynamic analysis (differential scanning calorimeter, DSC) showed that the heat of fusion ($\Delta H_m$) of the PVA-DOPA-Mucin blend increased with an increasing ratio of DOPA, as depicted in Supplementary Table 3. This increasing trend in the $\Delta H_m$ value could be attributed to the macromolecular interactions (H-bonds or entanglement of chains) between PVA-DOPA and mucin, and the interaction was strengthened with increasing DOPA content. Small-angle X-ray scattering (SAXS) was also performed, and the scattering profile of the PVA-DOPA-Mucin blend became increasingly distinct with an increasing DOPA ratio, which demonstrated that the interactions between PVA-DOPA and mucin were strengthened as the DOPA content increased and thus changed the overall conformational state of the PVA-DOPA chain, especially at higher concentrations (Fig. 3e)[42].

Therefore, from the results above, it could be speculated that the mucoadhesive property of the PVA-DOPA film could be partly attributed to the interpenetration and entanglement of polymer chains with the mucus. In addition to physical association, the mucoadhesion of the PVA-DOPA film was also due to the formation of hydrogen bonds and covalent bonds between the film and mucin. That is, catechol contains numerous hydroxyl groups that could form hydrogen-bonding groups. In addition, catechol forms quinone quickly upon oxidation, which can further react with amino or thiol groups found in the mucus layer, extracellular matrix proteins, or carbohydrates of mucus[8–10], thus confirming two theories of adhesion: the diffusion theory and the adsorption theory (Fig. 3f)[43].

**Synthesis and characterization of modified PLGA NPs.** To achieve a controlled drug release profile and improved mucus-penetrating ability and cellular uptake of the drug delivery system, the bare PLGA cores were modified with PEG, PVA, and PDA to obtain the final core-shell PLGA NPs (Fig. 1a). Figure 4a and Supplementary Fig. 4a-c clearly show that the prepared PLGA NPs had a uniform morphology and distribution. In addition, the core-shell structure of PLGA NPs with different surface modifications was observed in transmission electron microscopy (TEM) images (Fig. 4b). A PLGA core structure (diameter ~200 nm) could be observed in the center, with dense PEG, PVA, or PDA shells coated on the PLGA nanostructure. The sizes of the four PLGA NPs were evenly distributed, and the average particle sizes were 203.2 ± 12.7 nm, 221.7 ± 15.4 nm, 255.5 ± 11.8 nm, and 242.4 ± 19.0 nm, as measured by dynamic light scattering (Supplementary Fig. 4d, e). In addition, zeta potential analysis demonstrated that

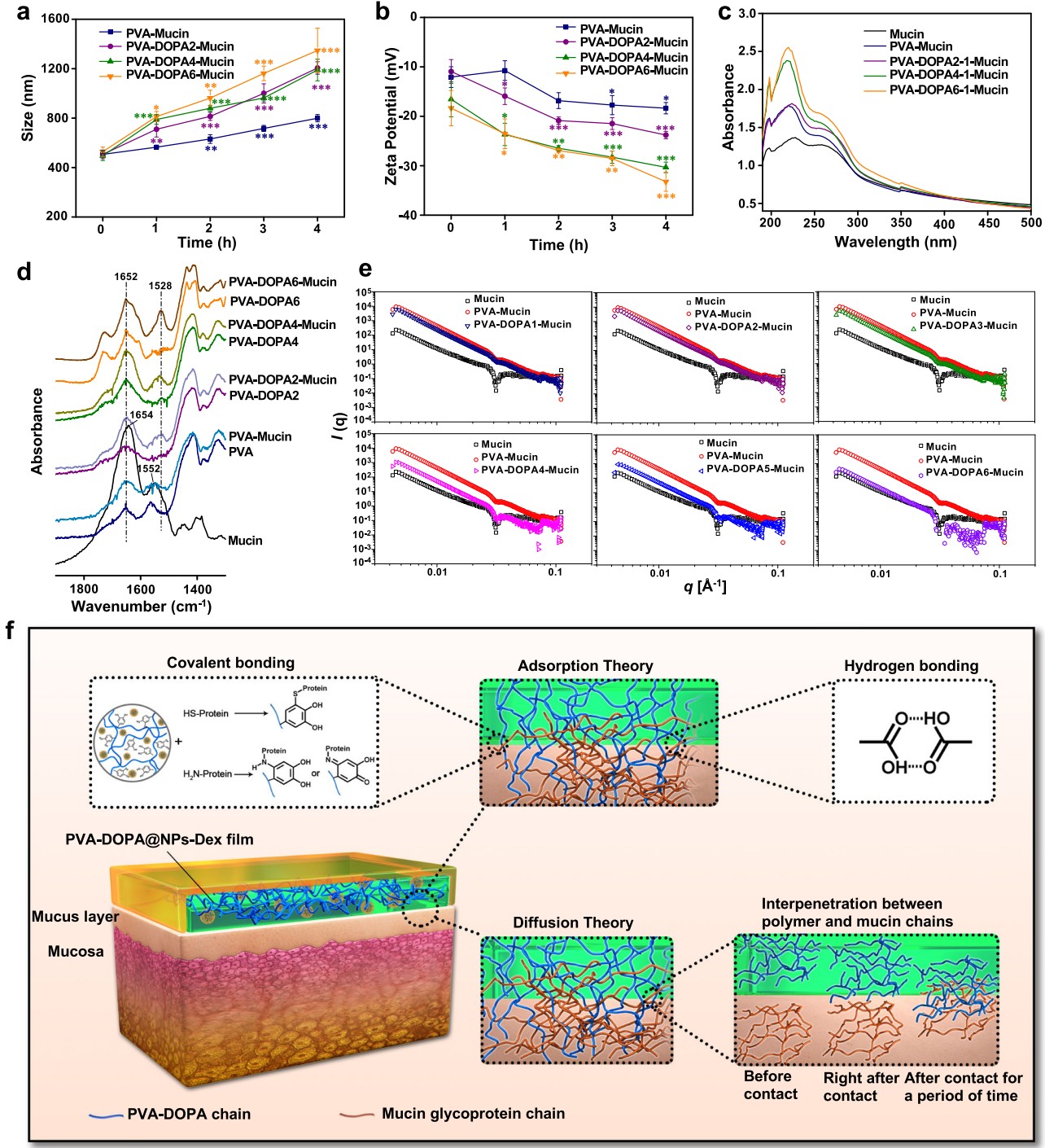

**Fig. 3 Interactions of PVA-DOPA films with mucin. a** Variation in the particle size of different PVA-DOPA-Mucin mixtures as a function of time. PVA: poly (vinyl alcohol), DOPA: 3,4-dihydroxy-D-phenylalanine. $n = 3$ independent samples per group; *$P = 0.025$; **$P < 0.01$; ***$P < 0.001$ vs value at 0 h. **b** Variation in the zeta potential of different PVA-DOPA-Mucin mixtures as a function of time. $n = 3$ independent samples per group; *$P < 0.05$; **$P < 0.01$; ***$P < 0.001$ vs value at 0 h. **c** UV-vis absorbance spectra of different PVA-DOPA-Mucin mixtures. **d** FTIR spectra of different PVA-DOPA before and after mixed with mucin. **e** SAXS spectra of different PVA-DOPA before and after mixed with mucin. **f** Schematic overview of the interactions between the PVA-DOPA film and mucus. NPs: nanoparticles, Dex: dexamethasone. All data are Mean ± S.D. Statistics was calculated by one-way ANOVA followed by Tukey's post-test. Exact $P$ values are given in the Source Data file. Source data are provided as a Source data file.

compared to PLGA NPs ($-15.4 \pm 0.7$ mV), PLGA-PEG NPs had a more neutral value of $-6.2 \pm 1.7$ mV, while PLGA-PDA NPs had a more negative surface charge ($-24.4 \pm 2.8$ mV) after PDA coating (Supplementary Fig. 4f) ($P < 0.01$). Then, the surface hydrophobicity of different PLGA NPs was determined using the Rose Bengal (RB) adsorption assay, and the results showed that

modified PLGA NPs had more hydrophilic properties, whereas the bare PLGA NPs were slightly more hydrophobic (Supplementary Fig. 4g), which can likely be attributed to the abundant hydroxyl groups introduced after surface modification of PLGA NPs. In summary, the results above clearly illustrated the successful surface decoration of different PLGA NPs.

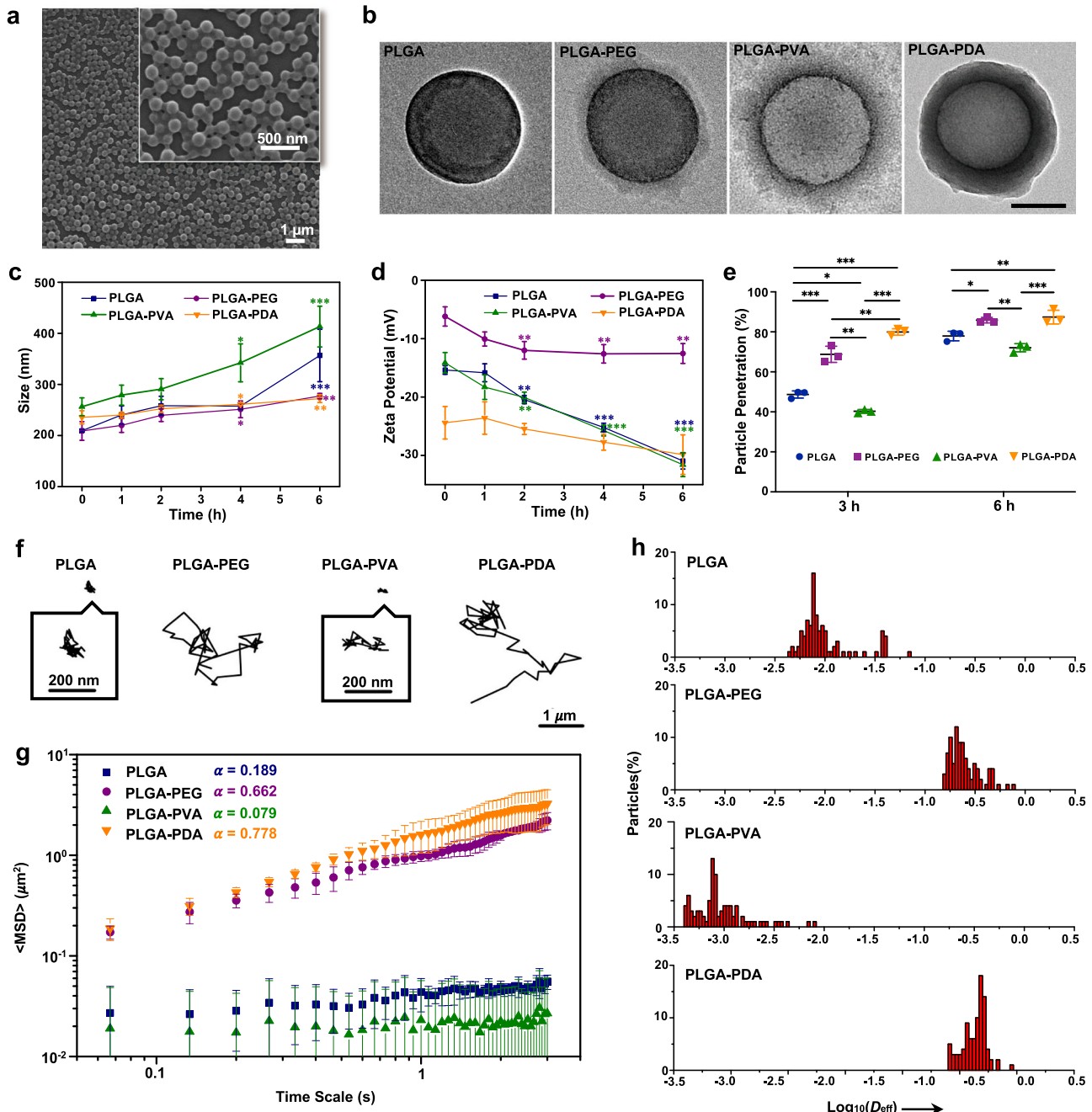

**Fig. 4 Characterization and mucus-penetrating properties of NPs in vitro. a** SEM image of PLGA NPs. **b** TEM images of PLGA, PLGA-PEG, PLGA-PVA, and PLGA-PDA NPs. PLGA: poly(lactic-co-glycolic acid), PEG: poly(ethylene glycol), PVA: poly(vinyl alcohol), PDA: polydopamine. Scale bar: 100 nm. **c** Variation in the particle size of different NPs-Mucin mixtures as a function of time. *$P < 0.05$; **$P < 0.01$; ***$P < 0.001$ vs value at 0 h. **d** Variation in the zeta potential of different NPs-Mucin mixtures as a function of time. **$P < 0.01$; ***$P < 0.001$ vs value at 0 h. **e** Percentage of NPs that penetrated across the mucus layer in a Transwell assay after 3 h and 6 h. *$P < 0.05$; **$P < 0.01$; ***$P < 0.001$. **f** Representative trajectories of different NPs in mucus. **g** MSD (mean squared displacement) values as a function of time scale for different NPs in mucus. **h** Distributions of the logarithms of individual particle effective diffusivities (D$_{eff}$) values at a time scale of 1 s. All data are Mean ± S.D. $n = 3$ independent samples per group. Statistics was calculated by one-way ANOVA followed by Tukey's post-test. Exact $P$ values are given in the Source Data file. Source data are provided as a Source data file.

**Mucus-penetrating properties of NPs.** After demonstrating the successful synthesis of NPs, the mucus-penetrating ability of different PLGA NPs was investigated. The efficient penetration of NPs through the mucus layer requires minimal interaction between NPs and the mucin particles[44]. Here, we first examined the ability of different NPs to adsorb mucin by probing the variation in the average particle size and zeta potential of NPs[44,45]. When mixed with a mucin suspension, bare PLGA and

PLGA-PVA NPs showed a significantly larger increase in average size than PLGA-PEG and PLGA-PDA NPs (Fig. 4c), indicating much less interaction of PLGA-PEG and PLGA-PDA NPs with the mucin particles. Zeta potential measurements showed a trend in accordance with particle size; that is, the variations in the zeta potential of bare PLGA and PLGA-PVA NPs were more obvious than those of PLGA-PEG and PLGA-PDA NPs (Fig. 4d), which confirmed that bare PLGA and PLGA-PVA NPs are more likely

to bond with mucin. Likewise, turbidity and mucin absorption percentages were also measured to investigate the interaction between NPs and mucin[45–47] (Supplementary Fig. 5a, b), which indicated that PLGA-PEG and PLGA-PDA NPs interacted with mucin much less than did bare PLGA and PLGA-PVA NPs.

Then, the ability of different NPs to penetrate through the mucus layer was quantified by using a Transwell system and agarose gel[44,45]. As the results indicated, the bare PLGA and PLGA-PVA NPs that interacted easily with the mucin particles also showed minimal mucus penetration, whereas PLGA-PEG and PLGA-PDA NPs exhibited much higher translocation across the mucus layer (Fig. 4e and Supplementary Fig. 5c). To achieve direct observation of the NP distribution in the mucus layer, the diffusion process of NPs was monitored using 3D confocal laser scanning microscopy (CLSM) imaging[48]. Supplementary Fig. 5d represents the z-stacks of different NPs in mucin, and as shown, PLGA-PEG and PLGA-PDA NPs were found in deep layers along the z-direction, while bare PLGA and PLGA-PVA NPs were basically localized in the upper mucus layer, suggesting the potential of PLGA-PEG and PLGA-PDA NPs to penetrate through the mucus layer.

To further investigate the behavior of NPs in mucus, the trajectories of particles in mucus were analyzed using a multiple-particle tracking (MPT)[16,44,49]. It was observed that PLGA-PEG and PLGA-PDA NPs more readily to diffused across the mucus layer and spanned much larger distances, whereas bare PLGA and PLGA-PVA NPs exhibited highly constrained trajectories (Fig. 4f and Supplementary Movies 5–8). Then, the ensemble-averaged mean squared displacement (MSD) for different NPs was quantified and is shown in Fig. 4g. The rapid mobility of PLGA-PEG and PLGA-PDA NPs was reflected by their markedly higher MSD values than those of PLGA and PLGA-PVA NPs across all time scales. The slope of the MSD ($\alpha$) vs time scale curve on a log-log scale was also calculated to reflect the extent of impediment to particle diffusion ($\alpha = 1$ indicates unobstructed Brownian diffusion; $\alpha < 1$ suggests increasing impediment to diffusion as $\alpha$ decreases)[16]. The average $\alpha$ of the NPs (Fig. 4g) was consistent with the Brownian trajectories shown in Fig. 4f and indicated less hindered motion of PLGA-PEG and PLGA-PDA NPs. We also examined the distribution of the logarithms of the individual particle effective diffusivities ($D_{\mathrm{eff}}$) on a time scale of 1 s[49]. As depicted in Fig. 4h, almost all of the PLGA-PEG and PLGA-PDA NPs exhibited $D_{\mathrm{eff}}$ values >0.1 µm²/s, indicating a rapid diffusion rate across the mucus layer. In contrast, few PLGA and PLGA-PVA NPs exceeded that speed.

**Cellular transport of NPs in vitro.** In addition to the mucus layer, the epithelial layer, especially the lipid content of the buccal epithelium, also poses a great challenge for successful drug delivery[6,50]. Therefore, we next assessed the transport behavior of NPs across epithelial cells. For in vitro evaluation of the cellular uptake of NPs, we adopted human oral keratinocyte (HOK) and human gingival epithelial cell (HGEC) cell lines originating from the human oral mucosa. As shown, the PLGA-PDA NPs exhibited the highest cellular internalization among all tested samples in both cell lines (Fig. 5a, b and Supplementary Fig. 6a, b). The mean gray intensity level obtained from the fluorescence images also quantitatively confirmed significantly higher cellular uptake of PLGA-PDA NPs than other NPs ($P < 0.001$) (Fig. 5c, d). To further study the cellular internalization and localization of NPs, HOK treated with NPs for 2 h were visualized in TEM images. Notably, the PLGA-PDA NPs showed better cellular uptake capability than other NPs (Fig. 5e), and TEM images also showed that the NPs diffused into the oral mucosal epithelial cells by means of endocytosis (Supplementary Fig. 6c). In addition, an in vitro monolayer model including a mucus layer was

developed[51,52]. TR146 cells were adopted as a cell model because they could form an epithelium resembling that of the non-keratinized buccal mucosa[53]. A mucus layer was deposited onto the TR146 cell monolayers[51,52]. The TR146 cell monolayer model treated with PLGA-PDA NPs showed the most obvious fluorescence intensity (Fig. 5f and Supplementary Fig. 6d) and the highest percentage of particle penetration across the cell monolayer (Fig. 5g), indicating the effective permeation of PLGA-PDA NPs across the mucus layer and epithelial cells. In contrast, the fluorescence intensity of the other three kinds of NPs was much weaker, suggesting decreased cellular internalization.

**Fabrication of the PVA-DOPA@NPs film and nanoparticle release and entry into tissues ex vivo and in vivo.** Next, different PLGA NPs were incorporated into the PVA-DOPA film to form a combined buccal drug delivery system (PVA-DOPA@NPs film). Tensile tests were performed again using fresh porcine buccal mucosa to determine the mucoadhesive properties of the films after nanoparticle encapsulation, and the results indicated that there was no significant difference in the mean detachment force between PVA-DOPA and PVA-DOPA@PLGA-PDA films (Supplementary Fig. 7a), which confirmed that nanoparticle incorporation had no adverse effects on mucoadhesive strength. In addition, since the release of NPs from the film is a primary requisite for permeation through the mucosa and subsequent drug release, we then performed an in vitro NP release analysis of various PVA-DOPA@PLGA-PDA films with different DOPA contents. As presented in Supplementary Fig. 7b, PLGA-PDA NPs could be gradually released from all kinds of PVA-DOPA@PLGA-PDA films, and the release rate was basically proportional to the amount of DOPA due to the different erosion rates of various PVA-DOPA films, as measured previously.

Although it has been demonstrated that the PDA-modified NPs show satisfactory mucus-penetrating behavior and superior cellular uptake, NP release from the film and subsequent entry into tissues remain to be investigated. Therefore, we next applied different PVA-DOPA6@NPs films onto porcine and rat buccal tissue to examine the ex vivo and in vivo efficiency of entry into mucosal epithelial tissues for different NPs. Both the ex vivo and in vivo results confirmed that compared with other tested NPs, the PLGA-PDA NPs maintained their superior permeability in terms of transport across the epithelium (Fig. 6). Combined, these results verified the superiority of PDA-modified PLGA NPs in epithelial cellular uptake.

**Toxicity evaluation of PVA-DOPA@PLGA-PDA films in vitro and in vivo.** Preclinical safety assessment is one of the most important preconditions for drug delivery platforms before clinical translation[54,55]. Therefore, a series of toxicity studies, including in vitro cytotoxicity, in vivo histopathology of the buccal mucosa and major organs, hematological examination, and biochemical indexes, were performed[54,55]. The results of the CCK-8 assay and cell attachment analysis all suggested the high biocompatibility of the mucoadhesive film (Supplementary Fig. 8). In addition to in vitro cytotoxicity studies, in vivo biosafety was further evaluated in Sprague Dawley rats. We first evaluated the potential of the films to irritate the buccal mucosa. The results in Fig. 7a indicate that there was no significant inflammation, necrosis, or metaplasia in the buccal mucosa tissue in contact with the films for 4 h compared with the normal tissue. Moreover, further evaluation was performed to detect the histocompatibility of the film with major organs (the heart, liver, spleen, lung, and kidney). Both morphological observation (Supplementary Fig. 9a) and hematoxylin and erosion (H&E) staining results (Fig. 7b and Supplementary Fig. 9b) showed no

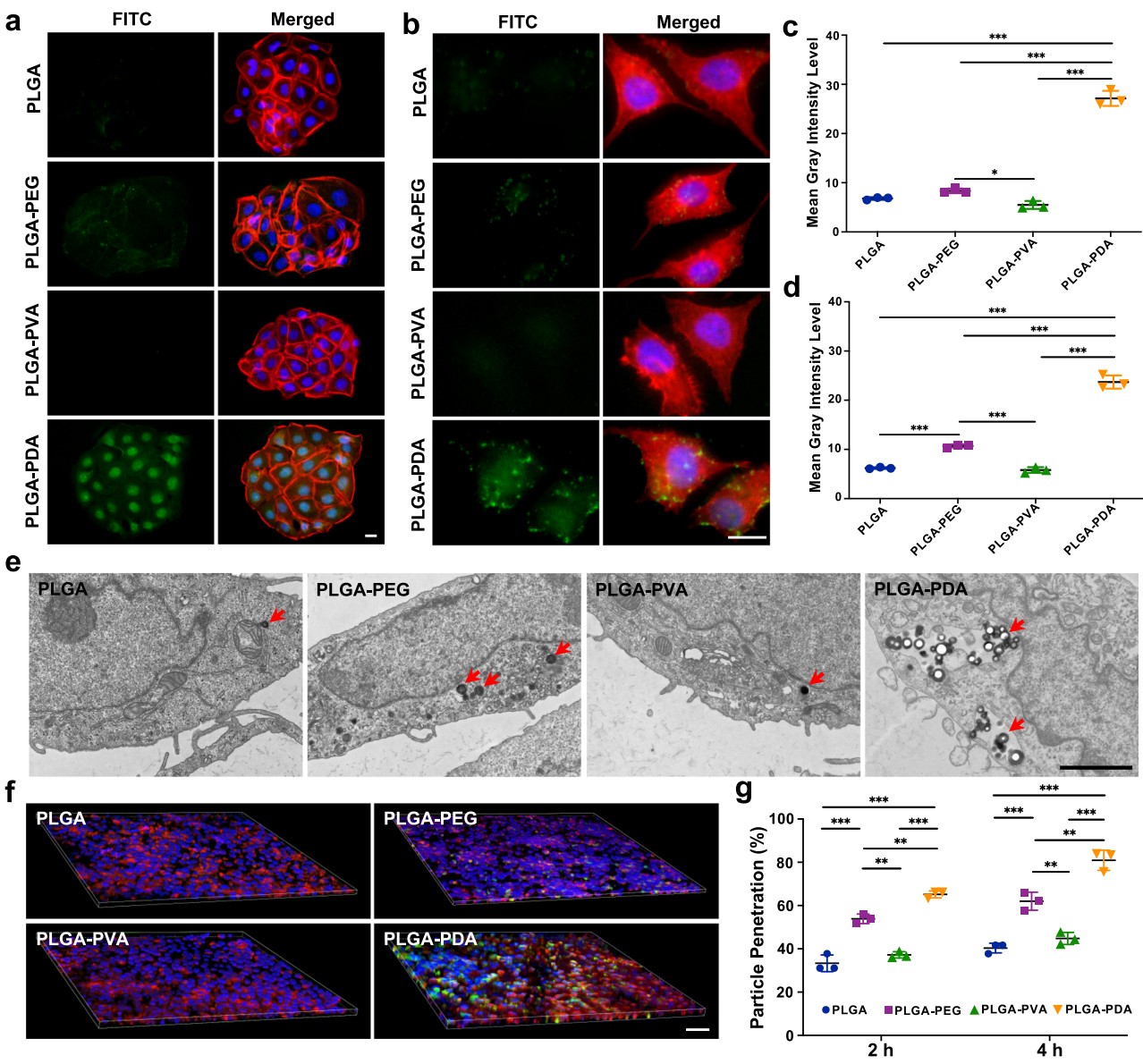

**Fig. 5 Cellular uptake of NPs in vitro. a, b** Fluorescence image of the cellular uptake of different NPs after incubation for 2 h in HOK and HGECs. Scale bars: 20 μm. PLGA: poly(lactic-co-glycolic acid), PEG: poly(ethylene glycol), PVA: poly(vinyl alcohol), PDA: polydopamine. **c, d** Quantification of the fluorescence intensity of different NPs obtained from fluorescence images of HOK and HGECs. *$P = 0.021$; ***$P < 0.001$. **e** TEM images of the cellular transport and localization of different NPs in HOK after incubation for 2 h. Scale bar: 2 μm. **f** 3D images of the cellular transport of NPs in the TR146 cell monolayer. Scale bar: 100 μm. **g** Percentage of NPs transported through the TR146 monolayer. **$P < 0.01$; ***$P < 0.001$. All data are Mean ± S.D. $n = 3$ independent cells per group. Statistics was calculated by one-way ANOVA followed by Tukey's post-test. Exact $P$ values are given in the Source Data file. Source data are provided as a Source data file.

obvious tissue damage or pathological change in any of the major organs following film administration for 1 or 7 days. In the hematological analysis, no changes were induced by the PVA-DOPA film administration (Fig. 7c–f). Finally, blood biochemistry was performed, and the results demonstrated that there was no significant change in the function of the heart ($Mg^{2+}$, $Ca^{2+}$, CK, and LDH-L), liver (ALT and AST) or kidney (BUN and CREA) after 1 or 7 days (Fig. 7g–n). Therefore, all of these toxicity assays demonstrated the high biosafety and biocompatibility of the PVA-DOPA@NP films, indicating their great potential for future clinical translation.

**In vitro drug release and in vivo pharmacokinetic study.** In the present study, dexamethasone (Dex), a widely used anti-

inflammatory drug, was chosen as a model compound to examine the possibility of using a mucoadhesive film for efficient buccal drug delivery[56–58]. The loading capacities of PLGA-Dex, PLGA-PEG-Dex, PLGA-PVA-Dex and PLGA-PDA-Dex NPs were 5.81 ± 2.38%, 8.68 ± 1.27%, 10.11 ± 1.49%, and 10.25 ± 0.98%, respectively. The in vitro release behavior of Dex from different PVA-DOPA6@NPs-Dex films was then evaluated, and it was clearly observed that all films with different PLGA NPs achieved a sustained drug release profile (Supplementary Fig. 10). However, the film with incorporated PLGA-PDA NPs showed more sustained drug release behavior than the other three NPs, which might be attributed to the dense PDA coating that formed outside the PLGA core (Fig. 4b) and delayed drug release from the NPs and the film.

In addition, we further studied in vivo pharmacokinetics in Sprague Dawley rats. The mean plasma concentration vs time

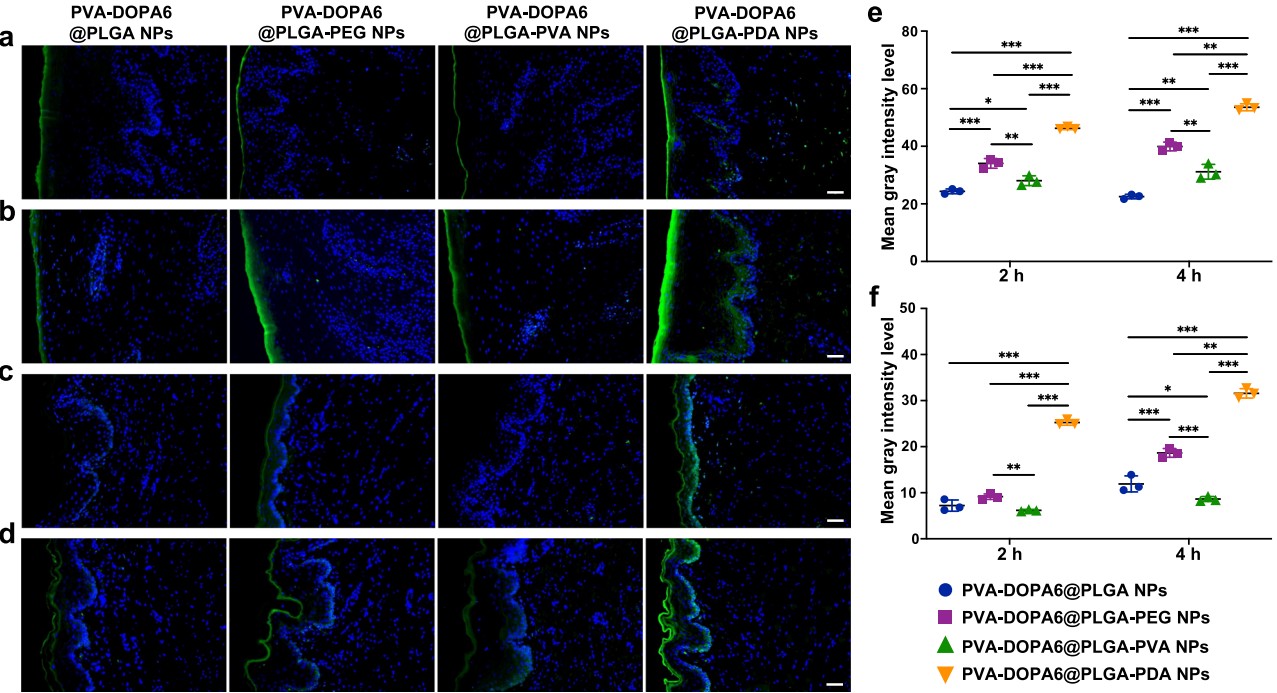

**Fig. 6 Ex vivo and in vivo permeation studies. a, b** Ex vivo distribution of different NPs in porcine buccal tissue incubated for 2 h and 4 h, respectively. Scale bars: 100 μm. PVA: poly(vinyl alcohol), DOPA: 3,4-dihydroxy-D-phenylalanine, PLGA: poly(lactic-co-glycolic acid), PEG: poly(ethylene glycol), PVA: poly(vinyl alcohol), PDA: polydopamine. **c, d** In vivo distribution of different NPs in rat buccal tissue incubated for 2 h and 4 h, respectively. Scale bars: 100 μm. **e** Quantification of the fluorescence intensity of different NPs obtained from fluorescence images of porcine buccal tissue. *$P = 0.031$; **$P < 0.01$; ***$P < 0.001$. **f** Quantification of the fluorescence intensity of different NPs obtained from fluorescence images of rat buccal tissue. *$P = 0.033$; **$P = 0.005$; ***$P < 0.001$. All data are Mean ± S.D. $n = 3$ animals per group. Statistics was calculated by one-way ANOVA followed by Tukey's post-test. Exact $P$ values are given in the Source Data file. Source data are provided as a Source data file.

profiles of different Dex formulations administrated via the oral or buccal routes are depicted in Fig. 8a, and the corresponding pharmacokinetic parameters are listed in Supplementary Table 4. The pharmacokinetic profiles of Dex were different between the two routes. The oral route resulted in a high maximum plasma concentration ($C_{max}$) at early time points, but the values also decreased rapidly and reached $T_{1/2}$. In contrast, the two kinds of buccal therapy provided a lower initial drug concentration but more sustained delivery of Dex from the prepared films, with $C_{max}$ values of 5.3 ± 2.3 h and 12.0 h and $T_{1/2}$ values of 8.9 ± 2.7 h and 20.7 ± 0.7 h, respectively. It was also observed that the film with embedded PLGA-PDA NPs achieved the highest area under the curve (AUC) value, which was 3.5-fold, 2.5-fold, and 2.1-fold greater than the values obtained for orally delivered Dex, orally delivered PLGA-PDA NPs, and the buccally delivered PVA-DOPA6@PLGA-Dex film, respectively. From the results above, it could be concluded that delivery of the drug via buccal application of the mucoadhesive film with incorporated drug-loaded NPs could enhance drug absorption efficiency compared to that of the oral route, possibly due to the direct transmucosal transport of the drug-loaded NPs into the systemic circulation. In addition, the superior mucus-penetrating and cellular transport abilities of the PDA-modified NPs can also improve drug bioavailability, allowing them to serve as an effective drug delivery nanocarrier.

**Therapeutic efficacy of the PVA-DOPA@NPs film in vivo.** Then, we compared the therapeutic effect of a commercial Dex ulcer film with different PVA-DOPA@NPs films loaded with Dex for the treatment of oral ulcers. Severe oral ulcers on the buccal mucosa of Sprague Dawley rats were treated with Kanghua Dex Film®, PVA@PLGA-PDA-Dex, PVA-DOPA6@PLGA-Dex, PVA-

DOPA6@PLGA-PDA-Dex film, or no treatment ($n = 3$, named groups 1–3 for $n = 1$–3). We observed ulcer sizes for 8 days (Fig. 8b, c, Supplementary Fig. 11, and Supplementary Fig. 12). The three kinds of PVA@NPs/PVA-DOPA@NPs films showed improved therapeutic effects compared with those of the commercially available Kanghua Dex Film®. In particular, PVA-DOPA6@PLGA-PDA-Dex film (91.51 ± 9.63%) was more effective in wound closure at day 5 than PVA@PLGA-PDA-Dex (58.21 ± 9.98%) or PVA-DOPA6@PLGA-Dex (62.95 ± 9.83%) ($P < 0.05$), again demonstrating the role of PVA-DOPA in prolonging the residence time of the film as well as the ability of PDA-modified NPs to mediate continuous delivery of the drug.

In histological analysis by H&E staining (Fig. 8d, Supplementary Figs. 13–15), the ulcers treated with PVA-DOPA6@PLGA-PDA-Dex film exhibited a completely regenerated epithelium similar to the normal buccal mucosa, whereas there was only partial healing for ulcers treated with other formulations or no treatment. Moreover, immunofluorescence staining for CK5 (expressed in the basal layer where proliferating cells are located) and CK13 (expressed in the intermediate layer and the parabasal layer)[59] showed complete coverage of the epithelium in the PVA-DOPA6@PLGA-PDA-Dex group, whereas coverage was incomplete in the other groups (Fig. 8d, Supplementary Fig. 14, and Supplementary Fig. 15). Finally, the inflammatory response was evaluated by CD11b staining, and the ulcers in all groups showed no obvious CD11b+ cells, demonstrating the biocompatibility of the administrated materials (Fig. 8d, Supplementary Fig. 14, and Supplementary Fig. 15).

## Discussion
In this study, we reported a biologically inspired mucoadhesive film combined with NPs for improved mucoadhesion and drug

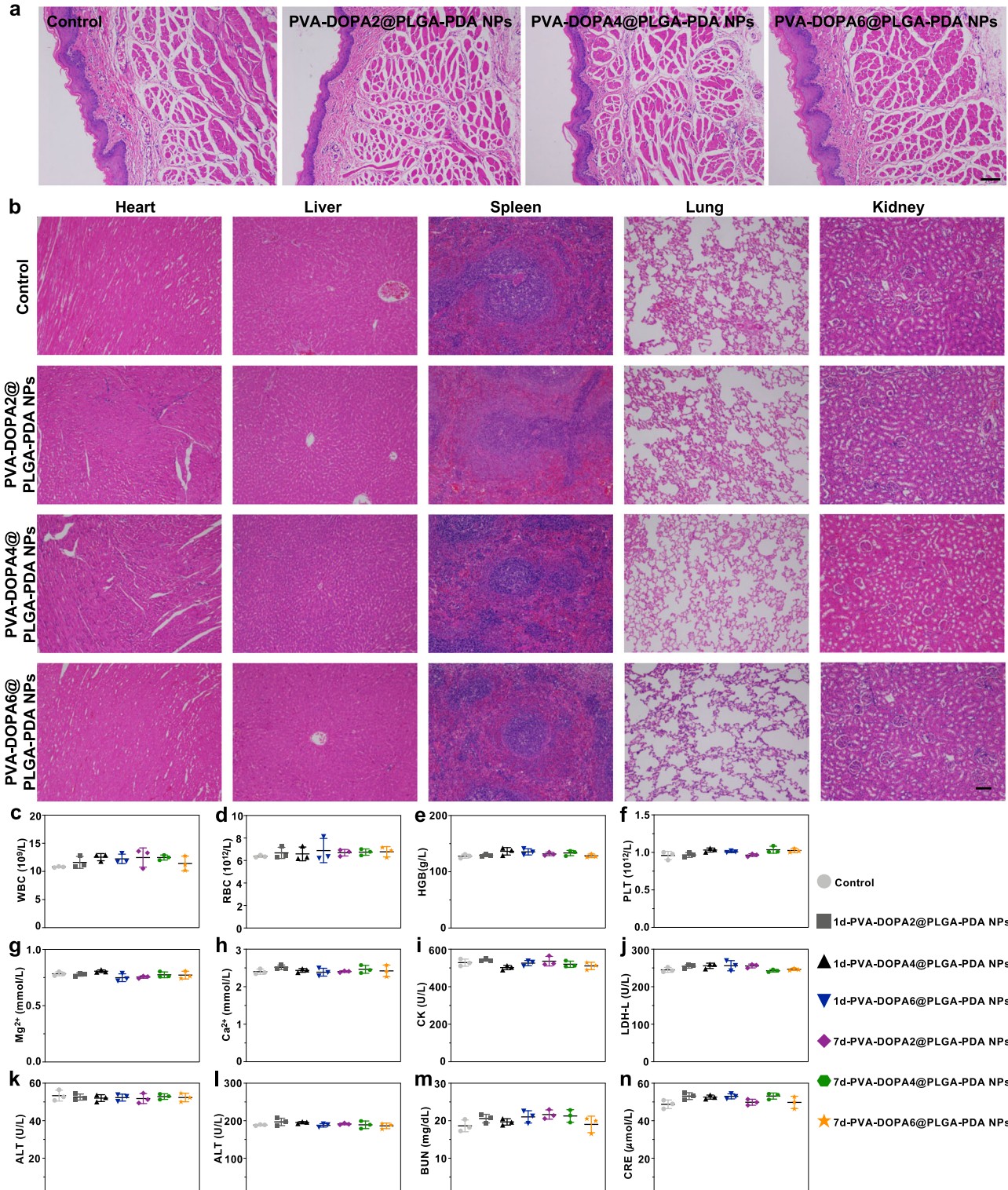

**Fig. 7 In vivo biosafety evaluation. a** Hematoxylin and erosion (H&E) staining of rat buccal mucosa tissue after application of different films for 4 h. PVA: poly(vinyl alcohol), DOPA: 3,4-dihydroxy-D-phenylalanine, PLGA: poly(lactic-co-glycolic acid), PDA: polydopamine. Scale bar: 100 μm. **b** H&E staining of major organs (the heart, liver, spleen, lung, and kidney) after subcutaneous implantation of different films in the backs of Sprague Dawley rats for 1 day. Scale bar: 100 μm. **c–f** Hematological examination of the variation in WBC (red blood cell count), RBC (white blood cell count), HGB (hemoglobin), and PLT (platelet count) after subcutaneous implantation of different films in the backs of Sprague Dawley rats for 1 or 7 days. **g–n** Blood biochemistry examination of the variation in $Mg^{2+}$, $Ca^{2+}$, CK (creatine kinase), LDH-L (lactate dehydrogenase), ALT (alanine transferase), AST (aspartate transferase), BUN (blood urea nitrogen), and CREA (creatinine) after subcutaneous implantation of different films in the backs of Sprague Dawley rats for 1 or 7 days. All data are Mean ± S.D. $n = 3$ animals per group. Statistics was calculated by one-way ANOVA followed by Tukey's post-test. Source data are provided as a Source data file.

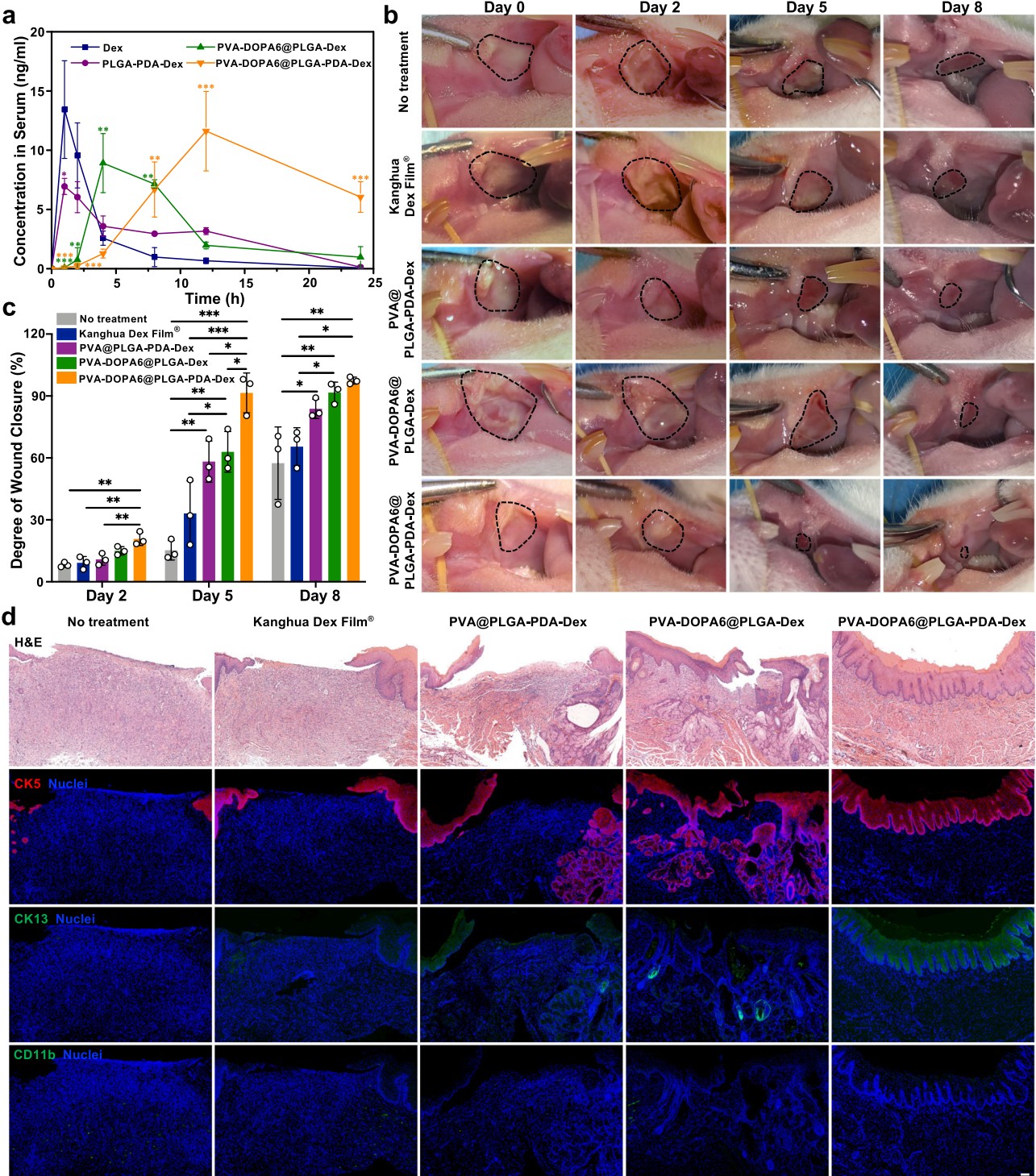

**Fig. 8 In vivo pharmacokinetic study and therapeutic efficacy of the PVA-DOPA@NP film in oral ulcers. a** Variation in plasma Dex concentration as a function of time after application of different formulations via the buccal or oral route. Dex: dexamethasone, PVA: poly(vinyl alcohol), DOPA: 3,4-dihydroxy-D-phenylalanine, PLGA: poly(lactic-co-glycolic acid), PDA: polydopamine. *$P = 0.022$; **$P < 0.01$; ***$P < 0.001$ vs Dex group. **b** Gross inspection of **b**uccal mucosa ulcers in Sprague Dawley rats treated with Kanghua Dex Film®, PVA@PLGA-PDA-Dex, PVA-DOPA6@PLGA-Dex, PVA-DOPA6@PLGA-PDA-Dex film, and no treatment at days 0, 2, 5 and 8 (Group 1). **c** Degree of wound closure of the oral ulcers at days 0, 2, 5 and 8. *$P < 0.05$; **$P < 0.01$; ***$P < 0.001$. **d** Hematoxylin and erosion (H&E) staining and immunohistochemical staining with anti-cytokeratin 5 rabbit monoclonal antibody (CK5, red), anti-cytokeratin 13 rabbit polyclonal antibody (CK13, green), and CD11b polyclonal antibody (CD11b, green) in regenerated oral ulcers at day 8. Nuclei (blue) were stained with DAPI (Group 1). Scale bar: 100 μm. All data are Mean ± S.D. $n = 3$ animals per group. Statistics was calculated by one-way ANOVA followed by Tukey's post-test. Exact $P$ values are given in the Source Data file. Source data are provided as a Source data file.

**Table 1 Summary of mucoadhesion strength, transport efficiency across mucosal barriers, drug bioavailability, and therapeutic efficacy of mussel-inspired PVA-DOPA film.**

**Adhesion strength on wet buccal tissue**

| Formulations | Lap-shear test (kPa) | Tensile test(kPa) | Peel Test(kPa) |
|---|---|---|---|
| Kanghua Dex Film® | 3.78 ± 0.56 | 2.85 ± 0.93 | 3.98 ± 1.35 |
| Zizhu Propolis Film® | 2.66 ± 0.20 | 1.94 ± 0.26 | 1.86 ± 0.37 |
| Yidebang Ulcer Film® | 6.15 ± 0.27 | 4.55 ± 0.47 | 3.95 ± 1.02 |
| Curatick® | 7.64 ± 1.29 | 5.23 ± 0.99 | 5.35 ± 0.72 |
| PVA | 1.66 ± 0.51 | 1.50 ± 0.03 | 1.62 ± 0.49 |
| PVA-DOPA1 | 3.59 ± 0.14 | 1.79 ± 0.45 | 3.53 ± 1.45 |
| PVA-DOPA2 | 4.42 ± 1.23 | 2.96 ± 0.87 | 4.76 ± 0.17 |
| PVA-DOPA3 | 7.67 ± 4.01 | 5.49 ± 3.10 | 5.62 ± 1.48 |
| PVA-DOPA4 | 11.46 ± 5.54 | 5.87 ± 1.16 | 7.83 ± 2.68 |
| PVA-DOPA5 | 28.81 ± 7.44 | 12.02 ± 1.43 | 13.51 ± 2.74 |
| PVA-DOPA6 | 38.72 ± 10.94 | 15.60 ± 4.11 | 15.82 ± 3.93 |
| **Transport efficiency across mucosal barriers** | | | |
| Formulations | TR146 cell monolayer model (penetration percentage) (%) | Pig ex vivo model (Mean gray intensity level) | SD rat in vivo model (Mean gray intensity level) |
| PVA-DOPA6@PLGA | 40.37 ± 2.25 | 22.53 ± 0.79 | 11.92 ± 1.73 |
| PVA-DOPA6@PLGA-PEG | 62.04 ± 4.17 | 39.93 ± 1.53 | 18.64 ± 0.94 |
| PVA-DOPA6@PLGA-PVA | 44.81 ± 2.80 | 31.15 ± 2.58 | 8.64 ± 0.54 |
| PVA-DOPA6@PLGA-PDA | 80.93 ± 4.66 | 53.55 ± 1.2 | 31.58 ± 1.03 |
| **Drug bioavailability** | | | |
| Formulations | $T_{max}$ (h) | $T_{1/2}$ (h) | $AUC_{0-24}$ (ng/ml*h) |
| Dex | 1.00 ± 0.0 | 2.1 ± 0.2 | 45.18 ± 11.48 |
| PLGA-PDA-Dex NPs | 1.00 ± 0.0 | 4.1 ± 2.7 | 64.70 ± 1.52 |
| PVA-DOPA6@PLGA-Dex | 5.3 ± 2.3 | 8.9 ± 2.7 | 78.12 ± 8.23 |
| PVA-DOPA6@PLGA-PDA-Dex | 12.0 ± 0.0 | 20.7 ± 0.7 | 160.17 ± 43.86 |
| **Therapeutic efficacy** | | | |
| Formulations | Degree of wound closure (%) | | |
| | Day 2 | Day 5 | Day 8 |
| Non-treatment | 8.20 ± 1.20 | 15.28 ± 4.76 | 57.44 ± 17.55 |
| Kanghua Dex Film® | 9.17 ± 3.05 | 33.13 ± 15.70 | 65.46 ± 9.18 |
| PVA@PLGA-PDA-Dex | 10.84 ± 2.83 | 58.21 ± 9.98 | 83.79 ± 4.68 |
| PVA-DOPA6@PLGA-Dex | 14.99 ± 2.03 | 62.95 ± 9.83 | 91.66 ± 5.05 |
| PVA-DOPA6@PLGA-PDA-Dex | 20.77 ± 3.18 | 91.51 ± 9.63 | 97.43 ± 1.70 |

Data are presented as the means ± standard deviations (SDs). ($n = 3$). $T_{max}$, time at which maximum plasma concentration is attained; $T_{1/2}$, elimination half-life. *AUC* area under concentration-time curve.

availability. This composite mucoadhesive film offers superb advantages over materials reported in previous studies: (1) strong adhesion in the wet environment of the oral cavity to enable an adequate residence time; (2) tunable size, thickness, and erosion rate in the form of a thin film to facilitate mechanical matching of tissues and potential applications in various kinds of diseases that demand different dosage intervals; (3) the ability to be transported across epithelial barriers; and (4) a controlled and prolonged drug release profile. Table 1 also summarizes the mucoadhesion strength, transport efficiency across mucosal barriers, drug bioavailability, and therapeutic efficacy of the mussel-inspired PVA-DOPA@NPs film, demonstrating its advantages and potential for application via the buccal route in other diseases.

As stated before, catechol groups play a key role in rapid adhesion and are prone to oxidize under oxidative or alkaline conditions. Thus, the PVA-DOPA polymer in the present study was synthesized under $N_2$ protection, and the film was formed by direct lyophilization and then stored under vacuum before application. The FTIR spectra (Supplementary 1a) first verified the formation of hydrogen-bonding interactions between the PVA matrix and catechol, which resulted in a decreased crosslinking density of the networks with increasing DOPA content, as evidenced by the rheological studies (Fig. 2c). This may be because the -OH and -NH$_2$ groups in the catechol groups formed hydrogen bonds with the -OH of PVA chains and thus increased the distances between PVA chains. In addition, the increased $G''$

(Fig. 2d) also demonstrated the noncovalent crosslinking of the PVA-DOPA film[29–32]. Then, we also demonstrated that most catechol groups on the PVA-DOPA films remained unoxidized by means of UV-vis and [1]H-NMR spectroscopy. As stated in previous studies, the absorbance peak at 280 nm in the UV-vis spectra was assigned to the catechol groups and revealed that the catechol groups were not oxidized[24,25]. In addition, the absorption peaks at approximately 305 and 400 nm belong to oxidized o-quinone[28,60]. Therefore, despite a slight shoulder peak at ~305 nm in the PVA-DOPA5 and PVA-DOPA6, most catechol groups in PVA-DOPA5 and PVA-DOPA6 remained unoxidized, as evidenced by the strong peak at 280 nm (Fig. 2a). The single absorbance peak at 280 nm for PVA-DOPA1-4 also reveals that the catechol groups of DOPA mostly remained unoxidized. In addition, since the extent of UV absorbance is directly related to the concentration of catechol groups, our results also demonstrated that the content of catechol increased from the PVA-DOPA1 film to the PVA-DOPA6 film (16.0–72.0 wt%) (Supplementary Table 1). Moreover, since [1]H-NMR spectroscopy could be used to track DOPA oxidation and crosslinking at the molecular level[24], the peak of the phenyl group in the aromatic region ($\delta = 6.62, 6.71, 6.78$) demonstrated that the different PVA-DOPA films obtained in the present study mostly remained unoxidized. In addition, we also quantified the number of catechol groups in PVA-DOPA to evaluate the potential mucoadhesive properties of the films using the [1]H-NMR results[24,25].

Using this method, the ratios of catechol conjugated to the PVA backbone were found to be ~4.7–64.6% (Supplementary Table 1), consistent with the present finding that the mucoadhesive strength increased with increasing DOPA content. In conclusion, it could be speculated that the different PVA-DOPA films obtained in the present study remained largely unoxidized and mostly did not covalently crosslink to form o-quinone or di-dopa crosslinks, indicating the stability of the PVA-DOPA films.

To achieve the mucus-penetrating ability of PLGA-PDA NPs, we took advantage of the hydrophilic and negative surface coating of PDA[44,61]. Therefore, the surface properties of PDA could enhance the mucus-penetrating ability of PDA coated PLGA NPs by minimizing interaction with the negatively charged and hydrophobic pockets in mucus. In addition, it has been stated in previous studies that NPs could be made mucus-penetrating or mucoadhesive after functionalization with the same polymer. For instance, depending on the molecular weight (Mw) of PEG, NPs can be made mucus-penetrating (when Mw is 2 kDa) or mucoadhesive (when Mw is 10 kDa)[21]. Moreover, densely grafted PEG of 10–40 kDa can also exhibit mucus-penetrating properties[62]. In the present study, the PDA chain was densely coated on the surface of PLGA with a thin layer (Fig. 4b), which reduced the entanglement of chains between PDA and mucin fibers. In addition, the negatively charged surface of PDA minimized charge interactions with the negative domains of mucins. In conclusion, the PLGA-PDA NPs diffuse rapidly through the mucus layer and exhibit mucus-penetrating properties. In addition, the enhanced cellular uptake of PDA-coated PLGA NPs could be attributed to the zwitterionic properties of PDA (iso-electric pH 4−4.5)[44,63]. In the lipid bilayer composed of glycerophospholipid molecules, the positively charged amino groups of PDA could interact with the negatively charged phosphate groups, while the negatively charged phenol groups on the PDA surface could interact with the positively charged choline groups on the lipid membrane[44]. Therefore, this combination of interactions with the cell membrane could enhance the cellular uptake of PLGA-PDA NPs.

In summary, this study presents the design of an effective mussel-inspired buccal film with incorporated PLGA-PDA-Dex NPs for improved residence time and mucoadhesion strength. Upon application onto the buccal tissue, the NPs are released gradually from the film and subsequently penetrate the mucus layer and translocate across the epithelium, followed by sustained drug release and improved therapeutic efficacy in treating oral mucositis. We anticipate that this platform could improve the efficiency of buccal drug delivery and inspire the rational design of tissue adhesives, wound dressings, and NP-based delivery systems in the near future.

## Methods

**Materials**. Poly(vinyl alcohol) (Mw 85–124 kDa), 3,4-dihydroxy-D-phenylalanine (DOPA), and poly(D,L-lactide-co-glycolide) (PLGA) were purchased from Sigma-Aldrich (USA). Ethyl cellulose, bovine submaxillary mucin, and poly(ethylene glycol) (PEG) were purchased from Shanghai Yuanye Bio-Technology Co., Ltd. (China). Lysozyme, dopamine hydrochloride, FITC, TRITC phalloidin, DAPI, Dex, RB sodium salt, Tris buffer, and agarose were purchased from Beijing Solarbio Science & Technology Co., Ltd. (China). Alexa Fluor-555-WGA was purchased from Thermo Fisher Scientific (USA). Enhanced cell counting kit-8 (CCK-8) was obtained from Saint-Bio Co., Ltd. (China). Anti-cytokeratin 5 rabbit monoclonal antibody (CK5, Abcam, Cat no. ab52635, Lot. GR3292032-7, dilution: 1:100), anti-cytokeratin13 rabbit polyclonal antibody (CK13, Servicebio, Cat no. GB11802, dilution: 1:500), and CD11b polyclonal antibody (CD 11b, Bioss, Cat no. bs-1014R, Lot. AG05216987, dilution: 1:100).

**Cell culture**. The HGEC, HOK and TR146 cell lines were purchased from GuangZhou Jennio Biotech Co., Ltd. (China). HOK, HGECs, and TR146 cells were maintained in minimum Eagle's medium (MEM; Gibco), keratinocyte serum-free medium (K-SFM; Thermo Fisher Scientific), and Roswell Park Memorial Institute 1640 medium (RPMI 1640; HyClone) supplemented with 10% fetal bovine serum

(FBS; Gibco) and 1% penicillin and streptomycin (100 IU/ml) at 37 °C under 5% $CO_2$, respectively.

**Animal care**. Male Sprague-Dawley rats aged 8 weeks (250 ± 10 g) were provided by Chengdu Dashuo Bio-Technology Co., Ltd. (China). All experiments involving animals were carried out in compliance with the Institutional Animal Care and Use Committee of Sichuan University, Chengdu, China. Rats were fed a standard laboratory diet with a 12 h/12 h light/dark cycle under SPF conditions and had at least 1 week of acclimatization before any animal experiment.

**Synthesis and characterization of the PVA-DOPA film**. The PVA-DOPA polymers were synthesized according to a previous study[28] with some modifications. Briefly, PVA (6 mmol) was dissolved in DMSO (30 ml) at 100 °C, and 0.75 g $NaHSO_4 \cdot H_2O$ was then added to the PVA solution. After the temperature decreased to 80 °C, different amounts of DOPA (1–6 mmol) ($n_{DOPA}$ : $n_{PVA}$ = 1:6, 1:5, 1:4, 1:3, 1:2, and 1:1) were added, which were represented by PVA-DOPA1, PVA-DOPA2, PVA-DOPA3, PVA-DOPA4, PVA-DOPA5, and PVA-DOPA6, respectively. Then, the reaction was kept at 80 °C for 24 h under $N_2$. After that, the solutions were purified by dialysis for 3 days using a dialysis membrane (MWCO 3,500 Da, Biosharp, China). The final product was freeze dried and stored under vacuum. An ethyl cellulose protective cover was then formed. Briefly, 0.5 ml of 4% ethyl cellulose ethanol solution was added into a 1.5 cm × 1.5 cm × 1 cm mold and dried under vacuum to form a square open cap with a diameter of 1.5 cm and a height of 1 mm. The PVA-DOPA film was prepared by pouring 5% w/v PVA-DOPA solutions (PVA-DOPA1-6) into the ethyl cellulose cap mold. Then, the mixtures were cured under vacuum at 37 °C. The synthesis of the PVA-DOPA polymers was confirmed by means of FTIR (Thermo Nicolet, USA), UV-vis (SHIMADZU UV-3600, Japan), and ${}^1$H-NMR (Bruker DRX, USA) spectroscopy. The morphology of the PVA-DOPA film was observed by SEM (FEI Hillsboro, USA), and the thickness of the film was measured by a digital screw gauge at five different locations ($n = 3$). Then, the films were immersed in distilled water for 30 min to measure the surface pH of the films using pH test strips. Tensile strength was measured using a universal testing machine (Instron 5567, USA) with a loading speed of 5 mm/min. The rheological properties of the film were characterized by a rotating rheometer (TA Instruments, USA) at 25 °C and a frequency sweep (0.01–100 Hz at 0.1 strain) experiment was carried out to examine the storage ($G'$) and loss ($G''$) modulus. For the hydration and degradation tests, the films were weighed ($W_0$) and immersed in PBS containing lysozyme (0.5% w/v). At regular intervals, the films were removed and weighed ($W_1$). Then, the swollen films were dried under vacuum overnight and weighed again ($W_2$) ($n = 3$). The percentage of hydration and the percentage of erosion was calculated using the following equations:

$$\text{Percentage of hydration(\%)} = \frac{(W_1 - W_0)}{W_0} \times 100\% \tag{1}$$

$$\text{Percentage of erosion(\%)} = \frac{(W_2 - W_0)}{W_0} \times 100\% \tag{2}$$

For the evaluation of mucoadhesion, the ex vivo residence time was first assessed by two methods. Fresh porcine buccal tissue obtained from a local slaughterhouse was glued onto a PTFE mold and glass slide, and the films were pressed for 10 s to attach to the mucosal tissue. For the flow-through method, a PTFE mold and a burette filled with PBS was used to simulate the flow of saliva at 0.5 ml/min[4]. For the rotating disc method, the glass slide was immersed in PBS in a beaker under magnetic stirring at 1000 rpm. At specific time intervals, the number of films that attached to the buccal tissue was recorded ($n = 6$). Then, lap shear and tensile strength were measured using a universal testing machine (Instron 5567, USA) with a loading speed of 10 mm/min. Finally, the self-healing property of the film was confirmed by fracture and reformation testing. In brief, the film was hydrated and cut into two pieces, and the two pieces were brought back into contact for 10 s, followed by stretching to observe the self-healing behavior of the film. For the in vivo mucoadhesion analysis, male Sprague Dawley rats aged 8 weeks (250 ± 10 g) were used. The rats were fasted overnight with free access to water and were anesthetized with 1% pentobarbital sodium (40 mg/kg). Then, different PVA-DOPA films ($n = 3$) were applied to the buccal mucosa of the rats. After administration for 4 h, the state of the films adhered to the buccal tissue was observed and recorded.

**Exploration of the interactions between the PVA-DOPA film and mucus.**
Bovine submaxillary mucin was dissolved in PBS (1 mg/ml) and sonicated for 30 min. The mucin suspension was then reacted with different PVA-DOPA solutions at 37 °C in a shaker (150 rpm). The size and zeta potential of the mixture were measured at specific time points using a Zetasizer Nano ZS90 (Malvern Instruments, UK) ($n = 3$). The turbidity of the mixtures was measured with a UV spectrophotometer (Thermo MK3, USA) at 600 nm ($n = 3$). The mixtures were examined by UV-vis, SAXS (Xenocs Xeuss 2.0, France), DSC (Mettler Toledo, Switzerland), and ${}^1$H-NMR (Bruker DRX, USA) spectroscopy.

**Preparation and characterization of modified PLGA NPs.** Forty milligrams of PLGA (Mw 7-17 kDa) were dissolved in 2 ml of acetonitrile and added to 40 ml of 1% PEG or PVA under sonication for 4 h. Then, the PLGA-PEG or PLGA-PVA NPs were collected by centrifugation at 12,000 g for 20 min, washed three times and resuspended in distilled water. The PLGA NPs were synthesized in the same way without the addition of PEG or PVA. To prepare PLGA-PDA NPs, the prepared PLGA NP solution was added to an alkaline 0.5 mg/ml dopamine hydrochloride solution (pH 10, adjusted with Tris buffer) and reacted for 6 h under gentle magnetic stirring. The PLGA-PDA NPs were then collected and purified by centrifugation three times at 20,000 g for 30 min. The fluorescent NPs were fabricated using the same method except for the addition of FITC in acetonitrile at the beginning of the process. Then, the fabrication and characteristics of the NPs were verified using inverted fluorescence microscopy (Leica, Germany), SEM, TEM, AFM, and a Zetasizer Nano ZS90. The surface hydrophobicity of NPs was evaluated using a RB assay. In brief, 200 μl of 1 mg/ml NP solution was mixed with 400 μl of 100 μg/ml RB solution and incubated for different time periods under magnetic stirring at 1000 rpm and 25 °C. Afterward, the mixtures were centrifuged at 12,000 g for 30 min, and the supernatant was read with a UV spectrophotometer at 550 nm (n = 3). The percentage of NPs interacting with RB solution was then calculated.

**Mucus permeation studies.** One milliliter of 1 mg/ml mucin suspension was mixed with 0.5 ml of 1 mg/ml PLGA, PLGA-PEG, PLGA-PVA, and PLGA-PDA NPs and sonicated for different time periods. Then, the size and zeta potential of the mixtures were measured using a Zetasizer Nano ZS90, and the turbidity was measured with a UV spectrophotometer[17]. The mucin-NP mixtures were then centrifuged at 12,000 g for 30 min, and the amount of unabsorbed mucin was detected with a UV spectrophotometer at 258 nm. The mixtures were also examined by UV-vis spectroscopy to explore the interactions between the mucin and NPs.

The mucus-penetrating ability of NPs was assessed using a Transwell system (0.4 μm pores, 24-well, Corning, USA). Briefly, 20 μl of 10 mg/ml mucin suspension was uniformly deposited on the Transwell insert. Then, 900 μl of ultrapure water was placed in the acceptor compartment, and 200 μl of 1 mg/ml FITC-labeled PLGA, PLGA-PEG, PLGA-PVA, and PLGA-PDA NPs were gently added to the donor compartment. The Transwell plate was incubated at 37 °C in a shaker (150 rpm) for 3 h and 6 h. Afterwards, 100 μl of the solution was removed from the acceptor chamber, and the percentage of permeated NPs was quantified using a UV spectrophotometer at a λ of 490 nm[17,44]. The results of the penetration test were confirmed by another method using agarose gel. In brief, 1 ml of agarose solution (0.3 w/v %) was dissolved at 100 °C, added to vials and hardened at room temperature. One milliliter of 10 mg/ml mucin suspension was then uniformly placed on the agarose gel, and 200 μl of 10 mg/ml concentrations of different nanoparticle solutions were placed on the mucus layer and incubated at 37 °C in a shaker (150 rpm). After 6 h, the NPs and the mucin suspension were removed, and the agarose gels were rinsed three times with distilled water, melted, and evaluated by UV spectrophotometry at 490 nm[45]. For 3D mucus penetration observations, the mucin suspension was stained with Alexa Fluor-555-WGA (10 μg/ml) for 10 min at 37 °C. Then, 1 ml of 20 mg/ml stained mucin was deposited into a confocal dish and placed on a shaker to obtain mucus layers of equal thickness. FITC-labelled NPs (100 μl, 1 mg/ml) were carefully added dropwise onto the mucus layer and incubated for 30 min at 37 °C[48]. Images were taken every 20 μm along the z-axis, and 3D images were generated using a CLSM (Nikon N-SIM, Japan).

**Multi-particle tracking (MPT).** Particle transport rates were analyzed by exploring the trajectories of FITC-labelled NPs. Different NPs (10 μl; 1 mg/ml) were added to the mucin suspension and incubated for 2 h at 37 °C. At least three independent experiments were carried out for each kind of NPs, and the trajectories of n ≥ 100 particles were determined for each experiment. Movies were captured at a temporal resolution of 66.7 ms for 20 s using a Leica fluorescence microscope with a tracking resolution of 10 nm. The trajectories of the NPs were then analyzed with ImageJ software. The time-averaged MSD and the Deff was calculated using the following equations:

$$MSD = [x(t + \tau) - x(t)]^2 + [y(t + \tau) - y(t)]^2 \quad (3)$$

$$D_{eff} = \frac{MSD}{4\tau} \quad (4)$$

Where x and y represent the NP coordinates at a certain time, and τ denotes the time scale[16,44,49].

**Cellular uptake of NPs.** To quantify the cellular uptake of different PLGA NPs, HOK and HGECs were seeded onto a 24-well plate at a density of 1 × 10^5 and incubated for 24 h. The cells were subsequently treated with FITC-labelled NPs (100 μl, 1 mg/ml for each well) for 4 h. Then, the cells were rinsed with PBS 2–3 times, and the F-actin and nuclei were stained with TRITC phalloidin and DAPI, respectively. After three rinses with PBS, the fluorescence intensity was observed with a Leica fluorescence microscope. The fluorescence intensities of NPs were quantified using ImageJ software by calculating the mean gray intensity level of each kind of NPs. In addition, the cell uptake efficiency was further confirmed by means of TEM. Briefly, HOK were seeded into a 6-well plate at a density of 5 × 10^5,

and incubated for 2 days and were then treated with NPs (500 μl 1 mg/ml for each well) for 2 h. Then, the cells were collected and fixed with 2.5% glutaraldehyde for TEM observation.

**Transepithelial transport study.** A transepithelial transport study of NPs was carried out on TR146 cells, which were seeded in 12-well Transwell plates (0.4 μm pores, Corning, USA) at a density of 2 × 10^4 cells/cm^2 and incubated for 27 days, with the mediums changed every other day. Transepithelial electrical resistance was measured with a Millicell ERS-2 electrical resistance meter (Millipore, USA) to monitor the integrity of the cell monolayer. Then, a mucin suspension was deposited onto the TR146 cell layer, and the cells were incubated at 37 °C for 24 h. On the day of the experiment, the cells were washed 2–3 times with HBSS and treated with 100 μl of 1 mg/ml of FITC-labelled NPs in HBSS. After incubation for 2 h and 4 h, 100 μl of samples was removed from the basolateral side, and the ratio of transported NPs was determined with a UV spectrophotometer. Subsequently, the cells seeded on the apical side were washed 2–3 times with HBSS and stained with TRITC phalloidin and DAPI. The membrane was then removed from the Transwell insert, mounted on a slide and observed by CLSM using the z-axial scanning to observe the transport efficiency of different NPs.

**Ex vivo and in vivo permeation studies.** Then, the prepared NPs were mixed with various PVA-DOPA solutions to form the PVA-DOPA@NPs film. The mucoadhesive properties of the film were confirmed again by means of tensile strength testing after incorporation of NPs. The release of FITC-labelled NPs from the films was also analyzed. Briefly, different films with various contents of DOPA were placed on the Transwell insert (0.4 μm pores,12-well). Then, 1.5 ml and 1 ml of ultrapure water was added to the acceptor compartment and the donor compartment, respectively. The Transwell plate was then incubated at 37 °C in a shaker (150 rpm) for 12 h. At regular intervals, 100 μl of the solution was removed from the acceptor chamber at certain time points, and the number of NPs released from the films was quantified using a UV spectrophotometer at a λ of 490 nm.

Then, the transport of NPs across the porcine buccal mucosa was monitored ex vivo. Briefly, films loaded with FITC-labelled NPs were applied to fresh porcine buccal tissue and removed after 3 h and 6 h (n = 3). The buccal tissues were fixed with 4% paraformaldehyde, sliced, stained with DAPI, and visualized under a fluorescence microscope (Leica, Germany). For quantitative analysis, ImageJ software was used to calculate the mean gray intensity level of each sample. Likewise, the in vivo absorption of NPs was investigated using male Sprague Dawley rats aged 8 weeks (250 ± 10 g), which were fasted overnight with free access to water before the experiment. The Sprague Dawley rats were anesthetized with 1% pentobarbital sodium (40 mg/kg). Subsequently, films loaded with FITC-labelled NPs were administrated to the buccal mucosa of rats (n = 3). After application for 2 h and 4 h, the rats were sacrificed, and the buccal tissue in contact with the film was collected. Then, the films were removed, and the tissues were treated as described for porcine tissue and observed under a Leica fluorescence microscope. The mean gray intensity level of each slice was also examined using ImageJ software.

**In vitro cytotoxicity assay.** HOK and HGECs were seeded into 24-well plates at a density of 1 × 10^5, and 1/4 of PVA-DOPA@PLGA-PDA films were added into each well. After cocultured for 1, 2, and 3 days, the relative cell viabilities were determined by a CCK-8 assay (n = 3). The cell attachment of the films was also investigated. HOK and HGECs were seeded into 24-well plates containing PVA-DOPA@PLGA-PDA films at a density of 1 × 10^5. After 24 h, the culture medium was removed and washed 2–3 times with PBS. Then, the cells were stained with TRITC phalloidin and DAPI or dehydrated with ethyl alcohol and observed under a fluorescence microscope or a scanning electron microscope, respectively.

**In vivo biosafety evaluation.** The in vivo biocompatibility of the films was examined using male Sprague Dawley rats aged 8 weeks (250 ± 10 g). The rats were fasted overnight with free access to water and were anesthetized with 1% pentobarbital sodium (40 mg/kg). Films with incorporated PLGA-PDA NPs (n = 3) were applied to the buccal mucosa of the rats for 4 h to assess the potential irritation to the buccal mucosa. In addition, films were subcutaneously implanted in the backs of Sprague Dawley rats for 1 or 7 days (n = 3). The animals were then euthanized and the buccal tissue or major organs (the heart, liver, spleen, lung, and kidney) were harvested, embedded in paraffin, sectioned, and stained with H&E for histological analysis. In addition, blood samples were obtained from the retro-orbital plexus for hematologic and biochemistry analysis.

**Preparation and characterization of Dex-loaded PLGA NPs.** Briefly, 20 mg of PLGA and 5 mg of Dex were dissolved in 1 ml of acetonitrile and added dropwise into 40 ml of 1% PEG or PVA under sonication for 4 h. Then, the PLGA-PEG-Dex or PLGA-PVA-Dex NPs were collected by centrifugation at 12,000 g for 20 min, washed three times and resuspended in distilled water. The PLGA-Dex NPs were synthesized in the same way without the addition of PEG or PVA. The Dex-loaded PLGA-PDA-Dex NPs were also synthesized using the same method, followed by PDA coating as described above. To measure loading capacity, the Dex-loaded NPs were lyophilized, weighed and dissolved in acetonitrile. Then, the concentration of

Dex was characterized by HPLC (Shimadzu LC-20AD, Kyoto, Japan) with an Ultimate Plus-C18 column (Welch, Shanghai, China) and a mobile phase consisting of acetonitrile/water (35/65 v/v) containing 0.1% trifluoroacetic acid (with a flow rate of 1 ml/min). Drug loading was calculated according to the following equation:

$$\text{Drug loading}(\%) = \frac{W_{\text{Dex}}}{W_{\text{NPs}}} \times 100\% \qquad (5)$$

where $W_{\text{Dex}}$ is the weight of Dex in NPs and $W_{\text{NPs}}$ denotes the weight of NPs. Then, the Dex-loaded NPs (125 µg of Dex) were dispersed into the PVA-DOPA films ($275 \pm 25$ mg) and the percentage of Dex in the films ranged from 0.04 to 0.05%.

**In vitro drug release**. Then, the in vitro drug release profile of Dex-loaded films was examined. PVA-DOPA6 films containing different Dex-loaded NPs were placed into a dialysis membrane (MWCO 3500 Da) and immersed in 50 ml of PBS at 37 °C with magnetic stirring at 100 rpm. At different time points, 1 ml of the solution was removed, and 1 ml of fresh PBS was added. The concentration of Dex released over time was detected using a UV spectrophotometer at 240 nm ($n = 3$).

**In vivo pharmacokinetics study**. Male Sprague Dawley rats aged 8 weeks ($250 \pm 10$ g) were fasted overnight with free access to water and were randomly divided into four groups ($n = 3$). (group A: Dex was orally administered; group B: PLGA-PDA-Dex NPs were orally administered; group C: PVA-DOPA6@PLGA-Dex film was applied via the buccal route; and group D: PVA-DOPA6@PLGA-PDA-Dex film was applied via the buccal route). The rats were anesthetized with 1% pentobarbital sodium (40 mg/kg), and all groups were administered a drug dose of 1 mg/kg. After 4 h, the films in group C and group D were removed. Blood samples were collected from the retro-orbital plexus at different time points (1, 2, 4, 8, 12, and 24 h) and centrifuged at 800 $g$ for 10 min to obtain the plasma. Then, the collected plasma samples were analyzed using liquid chromatography-mass spectrometry (Thermo Fisher Scientific, USA) to measure Dex levels. Pharmacokinetic parameters were calculated using DAS software (version 2.0).

**In vivo therapeutic efficiency**. Male Sprague Dawley rats aged 8 weeks ($250 \pm 10$ g) were anesthetized with 1% pentobarbital sodium (40 mg/kg), and oral ulcers were induced by placing a round filter paper ($5 \times 5$ mm) soaked with 70% acetic acid on the buccal mucosa for 3 min. Two days after inducing the oral ulcer (day 0), Kanghua Dex Film®, PVA@PLGA-PDA-Dex, PVA-DOPA6@PLGA-Dex, and PVA-DOPA6@PLGA-PDA-Dex films were applied onto the buccal mucosa ulcer, and the rats with no treatment were set as the blank control group ($n = 3$). The animals were treated with the same procedure at 2, 5, and 8 days after the first dressing, and gross observation was observed before each film application. The animals were sacrificed on day 8, and the buccal mucosa around the ulcer was collected, and the samples were harvested for histological and immunohistochemical analysis. CK5 and CK13 were used to evaluate the regeneration of the epithelium, and the anti-CD11b was adopted to evaluate the recruitment of inflammatory cells according to the manufacturer's instructions. The degree of wound closure was calculated using the following equation:

$$\text{Degree of wound closure}(\%) = \frac{A_x - A_0}{A_0} \times 100\% \qquad (6)$$

where $_{Ax}$ is the wound area at Day $x$ and $A_0$ is the Wound area at Day 0.

**Statistical analysis**. Data are presented as the means ± standard deviations (S.D). Data from experiments were analyzed with Origin 9.1. Statistical analysis was performed with statistical software (IBM SPSS Statistics, v22.0; IBM Corp). One-way analysis of variance (ANOVA) followed by Tukey's post hoc test was used for comparisons among multiple groups (*$P < 0.05$; **$P < 0.01$; ***$P < 0.001$).

**Reporting summary**. Further information on research design is available in the Nature Research Reporting Summary linked to this article.

## Data availability
The authors declare that all the relevant data supporting the findings of this study are available within the article and its Supplementary Information files and Supplementary Movies. Source data are provided with this paper.

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

## Acknowledgements

This work was financially supported by the China Postdoctoral Science Foundation (2020M683265, 2020M683228) and the National Natural Science Foundation of China (81970984, 81970985, 81771122, 82071164, 81901060, 81701031). We would like to thank Chao Huang, Weifeng Zhao, Zhipeng Gu, Huile Gao, Xiaomeng Gao, Siyu Qin, Mingxin Qiao, Yilin Mao for their help.

## Author contributions

S.H. and X.P. planned and executed the main experiments, analyzed the data, and wrote the main paper. L.D., Z.Z., Y.L., and J.C. performed some experiments and revised the paper. T.C. and P.J. analyzed the data and revised the paper. Q.W. and J.W. conceived and designed the experiments and revised the paper. All authors critically reviewed and approved the paper.

## Competing interests

The authors declare no competing interests.

## Additional information

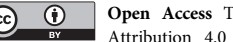

