## [Peer Review File · Nature Communications]

REVIEWER COMMENTS

Reviewer #1 (Remarks to the Author):

In this work, the authors designed catechol-bearing PVA/DOPA films incorporated with different PLGA NPs for buccal delivery. The nanoparticles and the films were carefully and thoroughly characterized both in vitro and in vivo. The findings also showed promising results. However, the manuscript needs to be further improved to suit to nature communications.

Major comments

1. A literature review on the previous works on mucoadhesive catechol-bearing materials should be added in the introduction. The novelty of the current work over the previous published articles should also be discussed and highlighted.
2. Since the title of the article is about mussel-inspired film, the authors should add more details on adhesive phenomenon of the mussel and the role of DOPA in the adhesion process in the introduction of the manuscript.
3. What was the technique used for the quantification of grafting percentage of DOPA in the film?
4. In the investigation of mucin-PVA/DOPA film interaction, please describe how does UV spectra indicate the covalent interaction occurred between DOPA and mucin. Would FTIR be more suitable for this investigation?
5. After film preparation, some catechol groups of DOPA were crosslinked together allowing the formation of covalently crosslink network of the films. However, only the free catechol groups that are employed for the mucoadhesion. Determining free catechol groups after film preparation may be essential as it will be directly related to the mucoadhesion properties of the films.
6. The authors conveyed that the extent of erosion increased as content of DOPA in the film was increased due to the self-crosslinking properties of DOPA molecules. I think self-crosslinking properties of DOPA molecules may be resulted in the decrease in mucoadhesive properties as the catechol moieties are already bind to themselves making to not available for the interaction with mucin. However, the film with the highest content of DOPA showed the highest mucoadhesion properties. The authors should discuss on this point in the results and discussion with reference support.
7. As reported by various articles, catechol is susceptible to oxidation leading to the change in the physical properties of the materials and may affect the adhesion properties of the films. The accelerated stability study of the prepared films should be investigated.
8. In the mucous permeation studies, what was the concentration and amount of mucin suspension used? Please specify in the manuscript.
9. In the exploration of the interaction mechanism between PVA-DOPA film and mucous, the authors mentioned that the UV absorbance increased with increasing DOPA ratio which could be assumed that PVA-DOPA formed catechol-mediated covalent bonding with mucin. I think the molecule of DOPA itself can also absorb UV which higher content of DOPA will also result in higher UV absorbance. The discussion on this aspect should not be discarded.
10. Various articles from literature reported that nanoparticles with higher mucoadhesive properties will stick on the mucin via the bond formation, and cannot penetrate to the deeper layer of mucin. Interestingly, the PLGA-PDA nanoparticles exhibited the highest penetration ability while providing the greatest covalent bond forming ability. This aspect should be discussed in the results and discussion with reference support.

11. Why did the PLGA-PDA NPs demonstrate significantly higher cellular uptake compared to other NPs? What could be the mechanism behind this finding? Please discuss this point in the results and discussion part.

12. Please provide more information on the loading of Dex into the NPs. How was the drug incorporated into the nanoparticles? And what was the loading content (loading capacity) of the drug in the nanoparticles? In the film preparation, what was the percentage of Dex in the film?

Minor comment

1. There are some typos and grammar mistake that need be to carefully checked. To make it appropriate for publication in nature communications, the English should be check by a native speaker or an English editing service.

2. Page 8 line 9, the term polymer should be used in place of film.

3. Page 11 line 3, radius or diameter?

4. The consistency of the abbreviation used should be checked throughout the manuscript

5. Page 19 line 21 "Dox" should be "Dex"

6. Page 21 'Discussion' seems to be 'conclusion'

Reviewer #2 (Remarks to the Author):

This manuscript focus on the development of a mussel-inspired film for adhesion of wet buccal tissue and efficient buccal drug delivery. The subject is of broad interest and the overall work is well planned and the findings are supported by the results. The work has a high quality and there are just two minor points to address:

1. There are several grammatical mistakes all over the text, so a review addressing that is advised.

2. It would be nice to describe in the abstract the specific main findings of the work.

Response to Review

We are grateful to the reviewers for their constructive comments and suggestions. Your comments are extremely important and helpful for us. We have carefully revised the paper according to the reviews' comments. Our point-by-point address to the comments is marked below in blue and the changes are also highlighted in the revised manuscript.

Reviewer #1 (Remarks to the Author):

In this work, the authors designed catechol-bearing PVA/DOPA films incorporated with different PLGA NPs for buccal delivery. The nanoparticles and the films were carefully and thoroughly characterized both in vitro and in vivo. The findings also showed promising results. However, the manuscript needs to be further improved to suit to nature communications.

Response: We thank the reviewer for his/her positive evaluation of our manuscript and constructive comments/suggestions, which have helped us improve the manuscript substantially. The detailed point-by-point responses are below and we have also revised the manuscript accordingly.

Major comments

1. A literature review on the previous works on mucoadhesive catechol-bearing materials should be added in the introduction. The novelty of the current work over the previous published articles should also be discussed and highlighted.

Response: We thank the reviewer for this suggestion. We have added a brief literature review about the previous works on mucoadhesive catechol-bearing materials in the introduction. The novelty of the current work over the previous published articles was also discussed and highlighted in both the introduction and discussion as follows:

“Inspired by this functional group from mussels, enormous efforts has been devoted to the development of catechol-functionalized adhesives via modifications of a variety of mucoadhesive polymers, including catechol-modified poly(ethylene glycol) (PEG)^{11,12}, chitosan^{5,13}, hyaluronic acid (HA)¹⁴, alginate¹⁵, etc. However, most of those studies focused on tissue adhesives for skin, and reports on formulations that exhibit excellent mucoadhesive properties in the wet oral environment and how they interact with the oral mucosa are still very limited. Although Xu et al.⁵ reported a catechol-chitosan mucoadhesive hydrogel for buccal drug delivery, they did not investigate the ability of the drug to be transported across the epithelial barrier, and their hydrogel

provided sustained drug release for only 3 h. Therefore, attempts should also be made to develop drug carriers that could be transported across the epithelial barriers with a controlled and prolonged drug release profile.

Recently, nanoparticles (NPs) have shown great promise for improved transport through the mucus barrier and can be tuned to support controlled or sustained release behaviour¹⁶⁻¹⁸. Among various strategies, surface modification of NPs with PEG has emerged as a popular strategy to enhance the mucus-penetrating ability of NPs¹⁹⁻²¹. Nevertheless, the hydrophilic and neutral surface properties of PEG may serve as a barrier for further cellular uptake of NPs²². Therefore, strategies that could achieve both excellent mucus-penetrating ability and cellular uptake across the epithelial barriers are also in high demanded.”

“In this study, we reported a biologically inspired mucoadhesive film combined with NPs for improved mucoadhesion and drug availability. This composite mucoadhesive film offers superb advantages over materials reported in previous studies: (1) strong adhesion in the wet environment of the oral cavity to enable an adequate residence time; (2) tunable size, thickness, and erosion rate in the form of a thin film to facilitate mechanical matching of tissues and potential applications in various kinds of diseases that demand different dosage intervals; (3) the ability to be transported across epithelial barriers; and (4) a controlled and prolonged drug release profile.”

2. Since the title of the article is about mussel-inspired film, the authors should add more details on adhesive phenomenon of the mussel and the role of DOPA in the adhesion process in the introduction of the manuscript.

Response: We thank the reviewer for the insightful comments. As the reviewer suggested, we have added more details on adhesive phenomenon of the mussel and the role of DOPA in the adhesion process in the introduction of the manuscript as follows:

“Marine mussels, which are well known for their remarkable underwater adhesion ability, have attracted widespread attention and are a potential source of an ideal tissue adhesive in the biological field^{8,9}. The rapid and robust adhesion of mussels could be attributed to the presence of the mussel adhesive proteins, which are abundant in the catecholic amino acid 3,4-dihydroxyphenylalanine (DOPA)^{8,9}. The catechol group of DOPA has an excellent maneuverability during crosslinking because it forms either covalent or noncovalent bonds. First, it can form noncovalent complexes, as in metal bidentate coordination and hydrogen bonding. In addition, the catechol groups oxidize easily to form o-quinone in oxidative or alkaline environments. The oxidized o-quinone is highly susceptible to forming covalent bonds with nucleophiles such as thiols and amines of proteins on the tissue surface via Michael addition or Schiff base reactions. In addition, o-quinone can also form di-dopa crosslinks via phenol radical coupling⁸⁻¹⁰.”

3. What was the technique used for the quantification of grafting percentage of DOPA in the film?

Response: We thank the reviewer for bringing this point to our attention. According to the results of $^1\text{H-NMR}$ spectra (Fig. R1) (Fig. 2b in the original manuscript), the grafting percentage of DOPA on the PVA-DOPA backbone was calculated by comparing the peak area of the phenyl group in catechol (a1-a3) ($\delta=6.62, 6.71, 6.78$) relative to that of methylene group in the PVA chains (b) ($\delta 1.5$) [J Am Chem Soc. 2017, 139(23): 8044-8050; Biomaterials. 2019, 216: 119268]. In the original manuscript, the authors meant to describe “the actual grafted DOPA/added DOPA” using “grafting percentage”. However, we realized that it is misleading, therefore, we calculated the degree of substitution of catechol in PVA-DOPA conjugates (the actual molar ratio of catechol/PVA-DOPA conjugates) according to the $^1\text{H-NMR}$ results using the same method. As shown in Supplementary Table 1, the degree of substitution of catechol ranged from 4.7% to 64.6%. Besides, the catechol content was also confirmed by the UV-vis spectroscopy, measuring absorbance at 280 nm (Fig. R2) (Fig. 2a in the original manuscript), and quantitative measurement was performed with a DOPA standard curve [Adv Funct Mater. 2020, 30(17):1910748; Small. 2019, 15(12):1900046]. As shown in Supplementary Table 1, the mass fraction of catechol/PVA-DOPA ranged from 16.0 wt% to 72.0 wt%. We regret that we did not clarify this information in the original manuscript and we have made relevant modifications in the revised manuscript as follows:

“The absorption peaks of catechol at 280 nm in the UV-vis spectra (Fig. 2a) and the peaks in the aromatic regions ($\delta= 6.62, 6.71, 6.78$) of the $^1\text{H-NMR}$ spectra (Fig. 2b) also verified the conjugation of DOPA to PVA chains^{24,25}, and the height of the absorption peaks was proportional to the amount of DOPA added. Therefore, the degree of substitution of catechol in PVA-DOPA conjugates could be calculated by comparing the peak area of the phenyl group in catechol ($\delta=6.62, 6.71, 6.78$) relative to that of the methylene group in the PVA chains ($\delta 1.5$)^{24,25}. As shown in Supplementary Table 1, the ratios of catechol conjugated to the PVA backbone were approximately 4.7-64.6%. Besides, the catechol content was confirmed by UV-vis spectroscopy, measuring absorbance at 280 nm, and quantitative measurement was performed with a DOPA standard curve^{26,27}. As shown in Supplementary Table 1, the mass fraction of catechol/PVA-DOPA ranged from 16.0 wt% to 72.0 wt%. Therefore, the catechol groups could be characterized and quantified by the UV-vis and $^1\text{H-NMR}$ spectroscopy.”

Figure R1. $^1\text{H-NMR}$ spectra of PVA-DOPA polymers with different content of DOPA.

Figure R2. UV-vis absorbance spectra of PVA-DOPA polymers with different amount of DOPA.

Supplementary Table 1. Degree of substitution of catechol and mass fraction of catechol in PVA-DOPA conjugates calculated from the results of $^1\text{H-NMR}$ and UV-vis spectra.

Samples	Molar ratio of PVA/DOPA	Degree of substitution of catechol calculated from $^1\text{H-NMR}$ (%)	Mass fraction of catechol calculated from UV-vis (wt%)
PVA-DOPA1	6:1	4.7	16.0
PVA-DOPA2	6:2	9.3	27.9
PVA-DOPA3	6:3	28.0	35.0
PVA-DOPA4	6:4	41.3	42.2
PVA-DOPA5	6:5	57.3	61.5
PVA-DOPA6	6:6	64.6	72.0

4. In the investigation of mucin-PVA/DOPA film interaction, please describe how does UV spectra indicate the covalent interaction occurred between DOPA and mucin. Would FTIR be more suitable for this investigation?

Response: We thank the reviewer for this comment. According to previous studies, the present study adopted the UV-vis spectra to investigate the covalent interaction

occurred between DOPA and mucin. Kim et al. monitored the covalent formation between catechol modified chitosan (Chi-C_{20.5}) and mucin by means of UV-vis spectra. They demonstrated that the slightly shifted absorption peak position of Chi-C_{20.5}-mucin complex was due to the covalent crosslinking between mucin and catechol [Biomaterials. 2015,52:161-70]. Besides, Pontremoli et al. and Yu et al. also demonstrated that the UV-vis spectroscopy could be used to investigate the formation of mucin-drug complex with a new structure [Bioorg Med Chem. 2015,23(20):6581-6; Spectrochim Acta A Mol Biomol Spectrosc. 2013, 103:125-9].

Therefore, the present study also used the UV-vis spectroscopy to investigate the interactions between mucin and PVA-DOPA. First, the absorption peaks of catechol in the UV-vis spectra (280 nm) [J Am Chem Soc. 2017, 139(23): 8044-8050; Biomaterials. 2019, 216: 119268] (Fig. R2) demonstrated the successful synthesis of different PVA-DOPA polymers. Then, different concentration of PVA or PVA-DOPA solutions were mixed with mucin. As shown, there was no obvious difference between mucin and PVA-mucin with different concentrations (Fig. R3a and R3b). However, a slight shift could be observed after mixing mucin with PVA-DOPA solutions (PVA-DOPA2, PVA-DOPA4, and PVA-DOPA6) (Fig. R3a). Besides, the absorption spectra of PVA-DOPA-Mucin complexes were also different from those of PVA-DOPA or mucin alone and the difference were larger with the increased concentration of PVA-DOPA (Fig. R3c-3e), indicating the covalent reaction of catechol with mucin. In conclusion, the catechol-mediated covalent reaction between different PVA-DOPA and mucin could be demonstrated by means of UV-vis spectra.

Thank the reviewer for this constructive advice. As you suggest, we have also investigated the interactions between PVA or PVA-DOPA and mucin using FTIR according to a previous study [Food Chem. 2020, 331:127355] and the results were shown in Fig. R4 below. The pure mucin exhibited amide I and amide II peak positions at 1645 and 1552 cm⁻¹, respectively (Fig. R4) and there was no significant difference between PVA and PVA-Mucin complex at the amide I and II bands. However, for the three kinds of PVA-DOPA polymers, amide II shifted to 1528 cm⁻¹ after reacting with mucin suspension. Therefore, the results of FTIR also demonstrated the covalent conjugation of catechol with mucin. We have also made relevant modifications in the revised manuscript as follows:

“First, the catechol-mediated covalent reaction between different PVA-DOPA and mucin was monitored by means of UV-vis spectroscopy^{13,39,40}. As shown in Fig. 3c and Supplementary Fig. 3b,

there was no obvious difference between mucin and PVA-mucin at different concentrations. However, a slight shift could be observed in the three kinds of PVA-DOPA-Mucin complexes. In addition, the absorption spectra of PVA-DOPA-Mucin complexes were also different from those of PVA-DOPA or mucin alone, and the difference was larger with an increased concentration of PVA-DOPA (Supplementary Fig. 3b), indicating the covalent reaction of catechol with mucin^{13,39,40}. We also investigated the interactions between mucin and PVA or PVA-DOPA using FTIR. The pure mucin exhibited amide I and amide II peak positions at 1645 and 1552 cm^{-1} , respectively (Fig. 3d), and there was no significant difference between PVA and the PVA-Mucin complex at the amide I and II bands. However, for the three kinds of PVA-DOPA, amide II shifted to 1528 cm^{-1} after reacting with mucin suspension, suggesting the covalent conjugation of catechol with mucin⁴¹.”

Figure R3. (a) UV-vis absorbance spectra of different PVA-DOPA-Mucin mixtures; (b-e) UV-vis absorbance spectra of different concentrations (0.05, 0.10, and 0.15 mg/ml) of PVA, PVA-DOPA2, PVA-DOPA4, and PVA-DOPA6 after mixed with mucin suspension.

Figure R4. FTIR spectra of PVA and PVA-DOPA before and after mixed with mucin suspension

5. After film preparation, some catechol groups of DOPA were crosslinked together allowing the formation of covalently crosslink network of the films. However, only the free catechol groups that are employed for the mucoadhesion. Determining free catechol groups after film preparation may be essential as it will be directly related to the mucoadhesion properties of the films.

Response: We appreciate the reviewer's valuable comments. First, we regret that we did not clarify the crosslinking status of the film in the original manuscript. The network of the PVA-DOPA films was formed mostly by the hydrogen-bond between PVA matrix and DOPA (the -OH and -NH₂ groups of catechol formed hydrogen bonds with the -OH of PVA chains), as evidenced by the FTIR results according to a previous study (an apparent shift of -OH from 3254 cm⁻¹ for pure PVA to 3277 cm⁻¹ with the addition of DOPA) (Fig. R5 below) [ChemSusChem. 2020,13(18):4974-84]. And we have also made relevant modifications in the revised manuscript.

Besides, we totally agree with you that the catechol groups play an essential role for mucoadhesion. As you suggest, we also quantified the catechol groups on PVA-DOPA conjugates to evaluate the potential mucoadhesion properties of the films by means of UV-vis and ¹H-NMR spectroscopy. As mentioned in Question 3, the catechol content was confirmed by the UV-vis spectroscopy, measuring absorbance at 280 nm (Fig. R2), and quantitative measurement was performed with a DOPA standard curve [Adv Funct Mater. 2020, 30(17):1910748; Small. 2019, 15(12):1900046]. As shown in

Supplementary Table 1, the mass fraction of catechol/PVA-DOPA ranged from 16.0 wt% to 72.0 wt%. Besides, according to the results of $^1\text{H-NMR}$ (Fig. R1), the degree of substitution of catechol in PVA-DOPA conjugates was also calculated by comparing the peak area of the phenyl group in catechol (a1-a3) ($\delta=6.62, 6.71, 6.78$) relative to that of methylene group in the PVA chains (b) ($\delta 1.5$) [J Am Chem Soc. 2017, 139(23): 8044-8050; Biomaterials. 2019, 216: 119268]. As shown Supplementary Table 1, the degree of substitution of catechol ranged from 4.7% to 64.6%. Therefore, the catechol groups could be characterized and quantified by the UV-vis and $^1\text{H-NMR}$ spectroscopy and we have also made relevant modifications in the revised manuscript as mentioned in Question 3.

Figure R5. FTIR spectra of PVA-DOPA polymers with different content of DOPA.

6. The authors conveyed that the extent of erosion increased as content of DOPA in the film was increased due to the self-crosslinking properties of DOPA molecules. I think self-crosslinking properties of DOPA molecules may be resulted in the decrease in mucoadhesive properties as the catechol moieties are already bind to themselves making to not available for the interaction with mucin. However, the film with the highest content of DOPA showed the highest mucoadhesion properties. The authors should discuss on this point in the results and discussion with reference support.

Response: We thank the reviewer for this insightful comment. We regret the inappropriate statement in the initial manuscript. Actually, the extent of erosion increased with the content of DOPA in the film was not due to the self-crosslinking properties of DOPA molecules, but due to the decreased crosslinking density of the network, as evidenced by the rheological results (G') (Fig. 2c in the original manuscript) [Biomed Mater. 2018, 13(2):025003]. This may be due to the fact the -OH and -NH₂ groups of catechol formed hydrogen bonds with the -OH of PVA chains and thus

increased the distances between molecules. We have also made relevant modifications in the revised manuscript.

In addition, we agree with you that the self-crosslinking properties of DOPA molecules may be resulted in the decrease in mucoadhesive properties. However, the different PVA-DOPA films obtained in the present study largely remained unoxidized and mostly did not covalently crosslinked to form o-quinone or di-dopa crosslinks the DOPA molecules. According to previous studies, the DOPA molecules tend to oxidized easily and form di-dopa crosslinks with each other under alkaline or oxidation condition [Chem Soc Rev. 2014, 43(24):8271-98; Mater Horiz. 2021, advance article]. However, the PVA-DOPA films in the present study were formed under neutral condition and lyophilized, thus the self-crosslinking of DOPA molecules was limited. In addition, we also proved this by means of UV-vis and $^1\text{H-NMR}$ spectra. The absorbance peak at 280 nm in the UV-vis spectra was assigned to the catechol groups and reveals that the catechol groups were not oxidized [J Am Chem Soc. 2017, 139(23): 8044-8050; Biomaterials. 2019, 216: 119268]. In addition, the absorption peaks at around 305 and 400 nm belongs to the oxidized o-quinone [Proc Natl Acad Sci USA. 2020,117(14):7613-21; Sci Transl Med. 2020,12(558):eaba8014.]. Therefore, despite a slight shoulder peak at around 305 nm in the PVA-DOPA5 and PVA-DOPA6, most catechol groups in PVA-DOPA5 and PVA-DOPA6 remained unoxidized, as evidenced by the strong peak at 280 nm (Fig. R2). Besides, the single absorbance peak at 280 nm of PVA-DOPA1-4 polymers also revealed that the catechol groups of DOPA mostly remained unoxidized. Furthermore, it has also been stated that the $^1\text{H-NMR}$ could be used to track DOPA oxidation and crosslinking in molecular level [J Am Chem Soc. 2017, 139(23): 8044-8050]. As shown in Fig. R6, the peaks in the aromatic region ($\delta=6.58, 6.68, 6.75$) indicated that the sample contained nonoxidized DOPA peptide. If the DOPA was oxidized, new peaks of o-quinone appeared ($\delta 6.19, 6.39, 7.09$) and the quinone peaks even disappeared because of the loss of aromatic hydrogens during cross-linking [J Am Chem Soc. 2017, 139(23): 8044-8050]. Therefore, since the $^1\text{H-NMR}$ results in the present study showed that there were peak areas of the phenyl group in catechol ($\delta=6.62, 6.71, 6.78$), but no obvious peak for o-quinone, it could be speculated that the different PVA-DOPA films obtained in the present study largely remained unoxidized (catechol group) and mostly did not covalently crosslinked to form o-quinone or di-dopa crosslinks.

Finally, as for the film with the highest content of DOPA showed the highest mucoadhesion properties in the present study, as mentioned in Question 5, the results of UV-vis and $^1\text{H-NMR}$ spectra also quantified the content of catechol groups, which increased with the added DOPA. Therefore, the film with the highest content of DOPA showed the highest mucoadhesion properties. We thank the reviewer for this constructive comment and we have now added discussions on this point in the results and discussion with reference support as follows:

“The absorption peaks of catechol at 280 nm in the UV-vis spectra (Fig. 2a) and the peaks in the aromatic regions ($\delta= 6.62, 6.71, 6.78$) of the $^1\text{H-NMR}$ spectra (Fig. 2b) also verified the conjugation of DOPA to PVA chains^{24,25}, and the height of the absorption peaks was proportional to the amount of DOPA added. Therefore, the degree of substitution of catechol in PVA-DOPA conjugates could be calculated by comparing the peak area of the phenyl group in catechol ($\delta=6.62, 6.71, 6.78$) relative to that of the methylene group in the PVA chains ($\delta 1.5$)^{24,25}. As shown in Supplementary Table 1, the ratios of catechol conjugated to the PVA backbone were approximately 4.7-64.6%. Besides, the catechol content was confirmed by UV-vis spectroscopy, measuring absorbance at 280 nm, and quantitative measurement was performed with a DOPA standard curve^{26,27}. As shown in Supplementary Table 1, the mass fraction of catechol/PVA-DOPA ranged from 16.0 wt% to 72.0 wt%. Therefore, the catechol groups could be characterized and quantified by the UV-vis and $^1\text{H-NMR}$ spectroscopy.”

“As stated before, catechol groups play a key role in rapid adhesion and are prone to oxidize under oxidative or alkaline conditions. Thus, the PVA-DOPA polymer in the present study was synthesized under N_2 protection, and the film was formed by direct lyophilization and then stored under vacuum before application. The FTIR spectra (Supplementary 1a) first verified the formation of hydrogen-bonding interactions between the PVA matrix and catechol, which resulted in a decreased crosslinking density of the networks with increasing DOPA content, as evidenced by the rheological studies (Fig. 2c). This may be because the -OH and -NH₂ groups in the catechol groups formed hydrogen bonds with the -OH of PVA chains and thus increased the distances between molecules. In addition, the increased G'' (Fig. 2d) also demonstrated the noncovalent crosslinking of the PVA-DOPA film²⁹⁻³². Then, we also demonstrated that most catechol groups on the PVA-DOPA films remained unoxidized by means of UV-vis and $^1\text{H-NMR}$ spectroscopy. As stated in previous studies, the absorbance peak at 280 nm in the UV-vis spectra was assigned to the catechol groups and revealed that the catechol groups were not oxidized^{24,25}. In addition, the absorption peaks at approximately 305 and 400 nm belong to oxidized o-quinone^{28,60}. Therefore, despite a slight shoulder peak at approximately 305 nm in the PVA-DOPA5 and PVA-DOPA6, most catechol groups in PVA-DOPA5 and PVA-DOPA6 remained unoxidized, as evidenced by the strong peak at 280 nm (Fig. 2a). The single absorbance peak at 280 nm for PVA-DOPA1-4 also reveals that the catechol groups of DOPA mostly remained unoxidized. In addition, since the extent of UV absorbance is

directly related to the concentration of catechol groups, our results also demonstrated that the content of catechol increased from the PVA-DOPA1 film to the PVA-DOPA6 film (16.0-72.0 wt%) (Supplementary Table 1). Moreover, since $^1\text{H-NMR}$ spectroscopy could be used to track DOPA oxidation and crosslinking at the molecular level²⁴, the peak of the phenyl group in the aromatic region ($\delta=6.62, 6.71, 6.78$) demonstrated that the different PVA-DOPA films obtained in the present study mostly remained unoxidized. In addition, we also quantified the number of catechol groups in PVA-DOPA to evaluate the potential mucoadhesive properties of the films using the $^1\text{H-NMR}$ results^{24,25}. Using this method, the ratios of catechol conjugated to the PVA backbone were found to be approximately 4.7-64.6% (Supplementary Table 1), consistent with the present finding that the mucoadhesive strength increased with increasing DOPA content. In conclusion, it could be speculated that the different PVA-DOPA films obtained in the present study remained largely unoxidized and mostly did not covalently crosslink to form o-quinone or di-dopa crosslinks, indicating the stability of the PVA-DOPA films.”

Figure R6. (A) Scheme showing oxidation of DOPA and possible resulting covalent crosslinks. (B) $^1\text{H NMR}$, DOPA, and ortho-quinone [J Am Chem Soc. 2017, 139(23): 8044-8050].

7. As reported by various articles, catechol is susceptible to oxidation leading to the change in the physical properties of the materials and may affect the adhesion properties of the films. The accelerated stability study of the prepared films should be investigated.

Response: We thank the reviewer for the suggestion. The PVA-DOPA polymer was synthesized under N_2 protection and the film was formed by direct lyophilization of PVA-DOPA solution. Therefore, we reduced the possibility of oxidation of catechol and the subsequent covalent crosslinking of the network directed by the oxidized o-quinone. In addition, as mentioned before, the $^1\text{H-NMR}$ and UV-vis results in the current study were obtained by dissolving the lyophilized PVA-DOPA film into D_2O and DD water,

respectively. And the results both demonstrated that the catechol groups in different PVA-DOPA films largely remained unoxidized (Fig. R1 and Fig. R2), thus proved the stability of the prepared films. In response to the reviewer's suggestion, we have also pointed it out in the revised manuscript.

8. In the mucous permeation studies, what was the concentration and amount of mucin suspension used? Please specify in the manuscript.

Response: We appreciate for the reviewer's careful comment. We have added the concentration and amount of mucin suspension and the relevant reference support in the mucus permeation studies in the revised manuscript.

9. In the exploration of the interaction mechanism between PVA-DOPA film and mucous, the authors mentioned that the UV absorbance increased with increasing DOPA ratio which could be assumed that PVA-DOPA formed catechol-mediated covalent bonding with mucin. I think the molecule of DOPA itself can also absorb UV which higher content of DOPA will also result in higher UV absorbance. The discussion on this aspect should not be discarded.

Response: Thank the reviewer for pointing out this issue. We totally agree with you that higher content of DOPA will result in higher UV absorbance and our results also demonstrated this phenomenon. As shown in Fig. R2, the higher the content of added DOPA, the stronger the absorbance peak at 280 nm in the UV-vis spectra.

As for how could the UV-vis absorbance be used to explore the interaction mechanism between PVA-DOPA film and mucus, the authors have explained in detail in Question 4. That is, after reacting with mucin suspension, a slight shift could be observed in PVA-DOPA-Mucin complexes (Fig. R3a). Besides, the absorption spectra of PVA-DOPA-mucin complexes were also different from those of PVA-DOPA or mucin alone and the difference were larger with the increased concentration of PVA-DOPA (Fig. R3c-3e), indicating the extent of reaction between PVA-DOPA and mucin [Biomaterials. 2015,52:161-70; Bioorg Med Chem. 2015,23(20):6581-6; Spectrochim Acta A Mol Biomol Spectrosc. 2013, 103:125-9]. Therefore, the catechol-mediated covalent reaction between different PVA-DOPA and mucin could be demonstrated by means of UV-vis spectra. We are sorry for the confusing statement in the original manuscript and we have added detailed explanation in the revised manuscript with reference support as mentioned in Question 4.

10. Various articles from literature reported that nanoparticles with higher mucoadhesive properties will stick on the mucin via the bond formation, and cannot penetrate to the deeper layer of mucin. Interestingly, the PLGA-PDA nanoparticles exhibited the highest penetration ability while providing the greatest covalent bond forming ability. This aspect should be discussed in the results and discussion with reference support.

Response: Thank the reviewer for making this insightful suggestion. Actually, the PLGA-PDA nanoparticles exhibited the highest penetration ability. However, the covalent bond forming ability with the mucosa was achieved by the PVA-DOPA polymer. The PLGA-PDA NPs were evenly dispersed in the PVA-DOPA network. Upon application onto the buccal tissue, the NPs release gradually from the film and subsequently penetrate the mucus layer and transport across the epithelium, followed by drug release.

To achieve mucus-penetrating ability of PLGA-PDA NPs, we take advantages of the hydrophilic and negative surface coating of polydopamine [ACS Appl Mater Inter. 2019,11(5):4777-89; Science. 2007, 318(5849):426-30]. This was likely due to the abundant hydroxyl groups present on PDA, which confer hydrophilicity, as well as the presence of other multiple functional groups (e.g., amino, phenol), which confer zwitterionic properties (isoelectric pH 4–4.5) [ACS Appl Mater Inter. 2019,11(5):4777-89; Chem Rev. 2014,114(9):5057-115]. Hence, at physiological pH, the phenolic groups deprotonate to a negative surface charge. Therefore, the surface properties of PDA could enhance the mucus-penetrating ability of PDA coated PLGA NPs by minimizing interaction with the negatively charged and hydrophobic pockets in mucus. Additionally, it has been stated in previous studies that the nanoparticles could be made mucus-penetrating or mucoadhesive after functionalized with the same polymer. For instance, depending on the molecular weight (Mw) of PEG, the PEG NPs can be made mucus-penetrating (when Mw is 2 kDa) or mucoadhesive (when Mw is 10 kDa) [Angew Chem Int Ed Engl. 2008,47(50):9726-9]. Moreover, the densely-grafted PEG of 10-40 kDa can also exhibit mucus-penetrating properties [Nanomedicine. 2016,11(11):1337-43].

Therefore, in the present study, the PDA layer was densely coated onto the PLGA surface with a thin layer (the thickness could be tuned by varying the polymerization time) [J Colloid Interface Sci. 2012, 386:366-72], which reduced the entanglement of

chains between PDA and mucin fibers. In addition, the negative charged surface of PDA minimized charge interactions with the negative domains of mucins. In conclusion, the PLGA-PDA NPs diffuse rapidly through the mucus layer and exhibit mucus-penetrating properties and we have added relevant information in the revised manuscript with reference support as follows:

“To achieve the mucus-penetrating ability of PLGA-PDA NPs, we took advantage of the hydrophilic and negative surface coating of polydopamine^{44,61}. Therefore, the surface properties of PDA could enhance the mucus-penetrating ability of PDA coated PLGA NPs by minimizing interaction with the negatively charged and hydrophobic pockets in mucus. Additionally, it has been stated in previous studies that nanoparticles could be made mucus-penetrating or mucoadhesive after functionalization with the same polymer. For instance, depending on the molecular weight (Mw) of PEG, NPs can be made mucus-penetrating (when Mw is 2 kDa) or mucoadhesive (when Mw is 10 kDa)²¹. Moreover, densely grafted PEG of 10-40 kDa can also exhibit mucus-penetrating properties⁶². Therefore, in the present study, the PDA chain was densely coated on the surface of PLGA with a thin layer, which reduced the entanglement of chains between PDA and mucin fibers. In addition, the negatively charged surface of PDA minimized charge interactions with the negative domains of mucins. In conclusion, the PLGA-PDA NPs diffuse rapidly through the mucus layer and exhibit mucus-penetrating properties.”

11. Why did the PLGA-PDA NPs demonstrate significantly higher cellular uptake compared to other NPs? What could be the mechanism behind this finding? Please discuss this point in the results and discussion part.

Response: We appreciate the reviewer for the valuable comments. As stated above, the presence of multiple functional groups (e.g., amino, phenol) endow PDA with zwitterionic properties (isoelectric pH 4–4.5) [ACS Appl Mater Inter. 2019,11(5):4777-89; Chem Rev. 2014,114(9):5057-5115]. Since the cellular membrane consisted of a lipid bilayer of glycerophospholipid molecules [ACS Appl Mater Inter. 2019,11(5): 4777-89; ACS Nano. 2013,7(10):9384-95], the enhanced cell uptake by PDA modified PLGA NPs could be attributed to its combination of positively charged amino groups, which promoted interaction with the negatively charged phosphate groups, and the negatively charged phenol groups on the PDA surface, which would interact with the positively charged choline groups on the lipid membrane [ACS Appl Mater Inter. 2019,11(5): 4777-89]. Collectively, these interactions with the cell membrane could promote the cellular uptake of PLGA-PDA NPs and we have provided relevant explanation in the revised manuscript as follows:

“Additionally, the enhanced cellular uptake of PDA-coated PLGA NPs could be attributed to the zwitterionic properties of PDA (isoelectric pH 4–4.5)^{44,63}. In the lipid bilayer composed of glycerophospholipid molecules, the positively charged amino groups of PDA could interact with the negatively charged phosphate groups, while the negatively charged phenol groups on the PDA surface could interact with the positively charged choline groups on the lipid membrane⁶¹. Therefore, this combination of interactions with the cell membrane could enhance the cellular uptake of PLGA-PDA NPs.”

12. Please provide more information on the loading of Dex into the NPs. How was the drug incorporated into the nanoparticles? And what was the loading content (loading capacity) of the drug in the nanoparticles? In the film preparation, what was the percentage of Dex in the film?

Response: We thank the reviewer for bringing up this issue. We are sorry not to report enough information on the loading of Dex into the NPs in the original manuscript and we have added relevant description in the revised manuscript as follows:

“Preparation and characterization of Dex-loaded PLGA NPs. Briefly, 20 mg of PLGA and 5 mg of Dex were dissolved in 1 ml of acetonitrile and added dropwise into 40 mL of 1% PEG or PVA under sonication for 4 h. Then, the PLGA-PEG-Dex or PLGA-PVA-Dex NPs were collected by centrifugation at 12,000 rcf for 20 min, washed 3 times and resuspended in distilled water. The PLGA-Dex NPs were synthesized in the same way without the addition of PEG or PVA. The Dex-loaded PLGA-PDA-Dex NPs were also synthesized using the same method, followed by PDA coating as described above. To measure loading capacity, the Dex-loaded NPs were lyophilized, weighed and dissolved in acetonitrile. Then, the concentration of Dex was characterized by HPLC (Shimadzu LC-20AD, Kyoto, Japan) with an Ultimate Plus-C18 column (Welch, Shanghai, China) and a mobile phase consisting of acetonitrile/water (35/65 v/v) containing 0.1% trifluoroacetic acid (with a flow rate of 1 ml/min). Drug loading (DL) was calculated according to the following equation: $DL (\%) = (\text{weight of Dex in NPs} / \text{weight of NPs}) \times 100\%$. Then, the Dex-loaded NPs (125 μg of Dex) were dispersed into the PVA-DOPA films (275 ± 25 mg) and the percentage of Dex in the films ranged from 0.04-0.05%.”

“The loading capacities of PLGA-Dex, PLGA-PEG-Dex, PLGA-PVA-Dex and PLGA-PDA-Dex NPs were $5.81 \pm 2.38\%$, $8.68 \pm 1.27\%$, $10.11 \pm 1.49\%$, and $10.25 \pm 0.98\%$, respectively.”

Minor comment

1. There are some typos and grammar mistake that need be to carefully checked. To make it appropriate for publication in nature communications, the English should be check by a native speaker or an English editing service.

Response: We thank the reviewer very much for this insightful suggestion. We have obtained the English Language Editing service from Nature Research Editing Service and the certificate is attached below:

2. Page 8 line 9, the term polymer should be used in place of film.

Response: We thank the reviewer for pointing out this typo, and we have replaced the term “polymer” with “film” in the revised manuscript

3. Page 11 line 3, radius or diameter?

Response: Many thanks for your reminder. We are sorry for the carelessness and we meant to write the term “diameter”. We have revised it in the revised manuscript.

4. The consistency of the abbreviation used should be checked throughout the manuscript

Response: We thank the reviewer for bringing up this issue. The authors have checked the consistency of the abbreviation throughout the manuscript and have made relevant modifications in the revised manuscript.

5. Page 19 line 21 “Dox” should be “Dex”

Response: Thank the reviewer for pointing out this typo. We are sorry for the

carelessness and we have replaced “Dox” with “Dex” in the revised manuscript.

6. Page 21 ‘Discussion’ seems to be ‘conclusion’

Response: We again thank the reviewer for these constructive comments. We have adopted the reviewer’s suggestion and have rewritten the “Discussion” section in the revised manuscript.

Reviewer #2 (Remarks to the Author):

This manuscript focus on the development of a mussel-inspired film for adhesion of wet buccal tissue and efficient buccal drug delivery. The subject is of broad interest and the overall work is well planned and the findings are supported by the results. The work has a high quality and there are just two minor points to address:

We truly appreciate the reviewer for the positive evaluation of our work and insightful comments. In accordance with the comments, we have revised the manuscript in a point-by-point manner.

1. There are several grammatical mistakes all over the text, so a review addressing that is advised.

Response: We thank the reviewer for the constructive suggestion. We have revised the manuscript and obtained the English Language Editing service from Nature Research Editing Service and the certificate is attached below.

This document certifies that the manuscript

A mussel-inspired film for adhesion of wet buccal tissue and efficient buccal drug delivery

prepared by the authors

Shanshan Hu, Xibo Pei, Lunliang Duan, Zhou Zhu, Yanhua Liu, Junyu Chen, Tao Chen, Ping Ji, Qianbing Wan & Jian Wang

was edited for proper English language, grammar, punctuation, spelling, and overall style by one or more of the highly qualified native English speaking editors at SNAS.

This certificate was issued on December 7, 2020 and may be verified on the SNAS website using the verification code 6C7B-CE31-DC92-7338-BEBP.

Neither the research content nor the authors' intentions were altered in any way during the editing process. Documents receiving this certification should be English-ready for publication; however, the author has the ability to accept or reject our suggestions and changes. To verify the final SNAS edited version, please visit our verification page at secure.authorservices.springernature.com/certificate/verify.

If you have any questions or concerns about this edited document, please contact SNAS at support@as.springernature.com.

SNAS provides a range of editing, translation, and manuscript services for researchers and publishers around the world. For more information about our company, services, and partner discounts, please visit authorservices.springernature.com.

2. It would be nice to describe in the abstract the specific main findings of the work.

Response: We would like to thank the Reviewer for pointing out this important point. According to your comment, we have revised the abstract and add more specific details about the finding of the work in the abstract. However, due to the word limit of abstract of Nature Communication (within 150 words), we are only able to add the main finding of the work in the abstract as follows:

“Administration of drugs via the buccal route has attracted much attention in recent years. However, developing systems with satisfactory adhesion under wet conditions and adequate drug bioavailability still remains a challenge. Here, we propose a mussel-inspired mucoadhesive film. Ex vivo models show that this film can achieve strong adhesion to wet buccal tissues (up to 38.72±10.94 kPa). We also demonstrate that the adhesion mechanism of this film relies on both physical association and covalent bonding between the film and mucus. Additionally, the film with incorporated polydopamine nanoparticles shows superior advantages for transport across the mucosal barrier, with improved drug bioavailability (~3.5-fold greater than observed with oral delivery) and therapeutic efficacy in oral mucositis models (~6.0-fold improvement in wound closure at day 5 compared with that observed with no treatment). We anticipate that this platform might aid the development of tissue adhesives and inspire the design of nanoparticle-based buccal delivery systems.”

References

1. Li, I. C. & Hartgerink, J. D. Covalent Capture of Aligned Self-Assembling Nanofibers. *J Am*

- Chem Soc* **139**, 8044-50 (2017).
2. Korupalli, C. *et al.* Single-injecting, bioinspired nanocomposite hydrogel that can recruit host immune cells in situ to elicit potent and long-lasting humoral immune responses. *Biomaterials* **216**, 119268 (2019).
 3. Zhao, X. *et al.* Physical Double-Network Hydrogel Adhesives with Rapid Shape Adaptability, Fast Self-Healing, Antioxidant and NIR/pH Stimulus-Responsiveness for Multidrug-Resistant Bacterial Infection and Removable Wound Dressing. *Adv Funct Mater* **30**, 1910748 (2020).
 4. Liang, Y. *et al.* Adhesive Hemostatic Conducting Injectable Composite Hydrogels with Sustained Drug Release and Photothermal Antibacterial Activity to Promote Full-Thickness Skin Regeneration During Wound Healing. *Small* **15**, 1900046 (2019).
 5. Kim, K., Kim, K., Ryu, J. H. & Lee, H. Chitosan-catechol: a polymer with long-lasting mucoadhesive properties. *Biomaterials* **52**, 161-70 (2015).
 6. Pontremoli, C., Barbero, N., Viscardi, G. & Visentin, S. Mucin-drugs interaction: The case of theophylline, prednisolone and cephalexin. *Bioorg Med Chem* **23**, 6581-6 (2015).
 7. Yu, X. *et al.* The investigation of the interaction between Oxymetazoline hydrochloride and mucin by spectroscopic approaches. *Spectrochim Acta A Mol Biomol Spectrosc* **103**, 125-9 (2013).
 8. He, W. *et al.* Covalent conjugation with (-)-epigallo-catechin 3-gallate and chlorogenic acid changes allergenicity and functional properties of Ara h1 from peanut. *Food Chem* **331**, 127355 (2020).
 9. Zhang, X. *et al.* Very Strong, Super-Tough, Antibacterial, and Biodegradable Polymeric Materials with Excellent UV-Blocking Performance. *ChemSusChem* **13**, 4974-84 (2020).
 10. Pinnaratip, R., Meng, H., Rajachar, R. M. & Lee, B. P. Effect of incorporating clustered silica nanoparticles on the performance and biocompatibility of catechol-containing PEG-based bioadhesive. *Biomed Mater* **13**, 025003 (2018).
 11. Yang, J., Cohen Stuart, M. A. & Kamperman, M. Jack of all trades: versatile catechol crosslinking mechanisms. *Chem Soc Rev* **43**, 8271-98 (2014).
 12. Zhang, X. *et al.* Polyphenol scaffolds in tissue engineering. *Mater Horiz* (2021) advance article.
 13. Priemel, T. *et al.* Compartmentalized processing of catechols during mussel byssus fabrication determines the destiny of DOPA. *Proc Natl Acad Sci USA* **117**, 7613-21 (2020).
 14. Xu, X. *et al.* Bioadhesive hydrogels demonstrating pH-independent and ultrafast gelation promote gastric ulcer healing in pigs. *Sci Transl Med* **12**, eaba8014 (2020).
 15. Poinard, B., Kamaluddin, S., Tan, A. Q. Q., Neoh, K. G. & Kah, J. C. Y. Polydopamine Coating Enhances Mucopenetration and Cell Uptake of Nanoparticles. *ACS Appl Mater Interfaces* **11**, 4777-89 (2019).
 16. Lee, H., Dellatore, S. M., Miller, W. M. & Messersmith, P. B. Mussel-inspired surface chemistry for multifunctional coatings. *Science* **318**, 426-30 (2007).
 17. Liu, Y., Ai, K. & Lu, L. Polydopamine and Its Derivative Materials: Synthesis and Promising Applications in Energy, Environmental, and Biomedical Fields. *Chem Rev* **114**, 5057-115 (2014).
 18. Wang, Y. Y. *et al.* Addressing the PEG mucoadhesivity paradox to engineer nanoparticles that "slip" through the human mucus barrier. *Angew Chem Int Ed Engl* **47**, 9726-9 (2008).
 19. Maisel, K. *et al.* Nanoparticles coated with high molecular weight PEG penetrate mucus and provide uniform vaginal and colorectal distribution in vivo. *Nanomedicine* **11**, 1337-43 (2016).
 20. Ball, V., Del Frari, D., Toniazzo, V. & Ruch, D. Kinetics of polydopamine film deposition as a function of pH and dopamine concentration: Insights in the polydopamine deposition mechanism. *J Colloid Interface Sci* **386**, 366-72 (2012).
 21. Liu, X. *et al.* Mussel-Inspired Polydopamine: A Biocompatible and Ultrastable Coating for Nanoparticles in Vivo. *ACS Nano* **7**, 9384-95 (2013).

REVIEWERS' COMMENTS

Reviewer #1 (Remarks to the Author):

The authors have addressed all of my concerns. The manuscript has been much improved and can be accepted for publication.